# DipM controls multiple autolysins and mediates a regulatory feedback loop promoting cell constriction in *Caulobacter crescentus*

Adrian Izquierdo-Martinez [1,2,9], Maria Billini[1,10], Vega Miguel-Ruano [3,10], Rogelio Hernández-Tamayo [4,5], Pia Richter[1], Jacob Biboy[6], María T. Batuecas[3], Timo Glatter [7], Waldemar Vollmer[6,8], Peter L. Graumann[4,5], Juan A. Hermoso [3] & Martin Thanbichler [1,2,5] ✉

Proteins with a catalytically inactive LytM-type endopeptidase domain are important regulators of cell wall-degrading enzymes in bacteria. Here, we study their representative DipM, a factor promoting cell division in *Caulobacter crescentus*. We show that the LytM domain of DipM interacts with multiple autolysins, including the soluble lytic transglycosylases SdpA and SdpB, the amidase AmiC and the putative carboxypeptidase CrbA, and stimulates the activities of SdpA and AmiC. Its crystal structure reveals a conserved groove, which is predicted to represent the docking site for autolysins by modeling studies. Mutations in this groove indeed abolish the function of DipM in vivo and its interaction with AmiC and SdpA in vitro. Notably, DipM and its targets SdpA and SdpB stimulate each other's recruitment to midcell, establishing a self-reinforcing cycle that gradually increases autolytic activity as cytokinesis progresses. DipM thus coordinates different peptidoglycan-remodeling pathways to ensure proper cell constriction and daughter cell separation.

In the course of evolution, cells have developed multiple strategies to reinforce their envelope in order to make it resistant to the internal osmotic pressure. Most bacterial species synthesize a semi-rigid cell wall surrounding the cytoplasmic membrane that bears part of the tension and, in addition, gives shape to the cells. The central component of the bacterial cell wall is peptidoglycan (PG)[1,2], a heteropolymer composed of glycan strands of alternating N-acetylglucosamine (NAG) and N-acetylmuramic acid (NAM) units that are covalently crosslinked by short peptide bridges[3]. The PG meshwork constitutes a single large macromolecule, the so-called sacculus, which needs to be constantly remodeled to enable cell growth, morphogenesis and cell division. This process requires the cleavage of bonds within the sacculus by lytic enzymes, also known as autolysins, and the subsequent insertion of new cell wall material by PG synthases. The activities of these two

[1]Department of Biology, University of Marburg, Marburg, Germany. [2]Max Planck Fellow Group Bacterial Cell Biology, Max Planck Institute for Terrestrial Microbiology, Marburg, Germany. [3]Department of Crystallography and Structural Biology, Instituto de Química-Física "Rocasolano", Consejo Superior de Investigaciones Científicas, Madrid, Spain. [4]Department of Chemistry, University of Marburg, Marburg, Germany. [5]Center for Synthetic Microbiology (SYN-MIKRO), Marburg, Germany. [6]Centre for Bacterial Cell Biology, Biosciences Institute, Newcastle University, Newcastle upon Tyne, UK. [7]Mass Spectrometry and Proteomics Facility, Max Planck Institute for Terrestrial Microbiology, Marburg, Germany. [8]Institute for Molecular Bioscience, The University of Queensland, Brisbane, QLD, Australia. [9]Present address: Bacterial Cell Biology, Instituto de Tecnologia Química e Biológica António Xavier, Universidade Nova de Lisboa, Oeiras, Portugal. [10]These authors contributed equally: Maria Billini, Vega Miguel-Ruano. ✉e-mail: thanbichler@uni-marburg.de

antagonistic groups of proteins need to be closely coordinated to prevent the emergence of weak spots in the PG layer that result in cell lysis[4].

Autolysins are a heterogeneous group of enzymes that are classified according to the bond they break in the PG molecule. Glycosidases and lytic transglycosylases (LTs) cleave the bonds between the sugar units of the glycan strands[5]. Notably, the reaction mediated by LTs produces 1,6-anhydro-NAM, which in some species acts as a signaling molecule indicating β-lactam antibiotic stress[6]. N-acetylmuramyl-L-alanine amidases (PG amidases) hydrolyze the bond between the L-alanine residue of the peptide and the lactyl moieties of NAM, generating naked glycan strands. They have been found to be required for daughter cell separation in various members of the gammaproteobacteria and firmicutes[7–10]. Finally, endopeptidases break various bonds in the peptide moieties, promoting PG incorporation and remodeling[11–14]. In general, individual autolysins are rarely essential. However, in many bacteria, the combined inactivation of multiple autolysins can cause strong morphological and/or synthetic lethal phenotypes[13,15–18].

The coordination of lytic and synthetic enzymes is thought to be achieved by their assembly into multi-protein complexes[4,19,20]. One of these complexes is the divisome, which carries out cell division in most bacteria[20,21]. Its assembly typically initiates with the polymerization of the tubulin homolog FtsZ into a dynamic ring-like structure at the future division site[22,23]. This so-called Z-ring then recruits, directly or indirectly, all other divisome components, including PG synthases, autolysins and regulatory proteins[24,25]. The coordinated activity of these factors gradually remodels the PG layer at the division site, constricting and ultimately splitting the sacculus to enable the release of the nascent daughter cells.

One of the most conserved mechanisms to regulate autolysin activity relies on the divisome components FtsE and FtsX, which form an ABC transporter-like complex in the cytoplasmic membrane[26]. It is thought that the ATPase activity of FtsE induces conformational changes in the transmembrane protein FtsX, which then conveys this signal to specific autolysins, thereby controlling their activity state[27–33]. In the case of gammaproteobacteria, this regulatory cascade involves endopeptidase homologs with a catalytically inactive LytM domain, so-called LytM regulators (also known as LytM factors), which act as adaptors linking FtsX to PG amidases[8,10,34–36]. However, there are also LytM regulators that stimulate amidase activity in an FtsEX-independent manner[8,10,35,37,38]. Their regulation and modes of action are highly variable among species, making it difficult to draw firm conclusions about the roles of LytM regulators in previously uncharacterized systems.

*Caulobacter crescentus* is a crescent-shaped Gram-negative bacterium that serves as one of the primary model organisms to study cell cycle regulation, cell differentiation and morphogenesis in the alphaproteobacteria[39–41]. It is characterized by a dimorphic life cycle in which a newborn motile swarmer differentiates into a sessile and division-competent stalked cell. The stalked cell then elongates and divides asymmetrically, giving rise to a swarmer and a stalked sibling[39]. During the swarmer cell stage, the cell body grows in length by disperse incorporation of new PG in the lateral cell walls. This process is mediated by the so-called elongasome, a multi-protein complex controlled by the actin homolog MreB[42,43]. After the initiation of divisome assembly, the *C. crescentus* cell switches to a medial mode of growth and, in addition, initiates stalk biosynthesis at its old pole[44–47]. Finally, activation of the divisome leads to progressive constriction of the midcell region, which culminates with the separation of the two nascent daughter cells.

Previous studies have comprehensively analyzed the autolytic machinery of *C. crescentus* and the contributions of its different components to cell morphogenesis in this species. A key regulatory factor is the LytM regulator DipM[48–50], a soluble periplasmic protein that carries two tandems of PG-binding LysM domains, and a C-terminal catalytically inactive LytM domain. The LysM domains are required for the accumulation of DipM at the division site, which initiates the early stages of divisome assembly[25,48–50]. Deletion of *dipM* causes pronounced cell filamentation, accompanied by membrane blebbing, although the published data do not completely agree on the severity of the phenotypic defects observed[48–50]. Interestingly, the LytM domain alone was reported to be sufficient to complement the Δ*dipM* phenotype, indicating that this domain most probably carries out the regulatory activity that DipM may have[48–50]. Previous work has shown that DipM is required, directly or indirectly, to recruit the putative soluble lytic transglycosylases (SLTs) SdpA and SdpB to the division site[18]. Moreover, in vitro studies revealed that it is able to stimulate the enzymatic activity of AmiC, the essential putative PG amidase of *C. crescentus*[18,51]. Notably, besides DipM, *C. crescentus* contains another soluble LytM regulator, named LdpF, which features two N-terminal coiled-coil domains and has a global architecture similar to that of *E. coli* EnvC[18]. LdpF is required for the midcell recruitment of AmiC, but in vitro evidence suggests that, in contrast to DipM, it is unable to stimulate the catalytic activity of AmiC[18,51]. Unlike the two LytM regulators, the catalytically active LytM endopeptidases (LdpA-E) of *C. crescentus* are largely redundant and only contribute to general cell fitness[18]. Notably, DipM and SdpA, together with the D,D-carboxypeptidase CrbA, are also recruited to the stalked pole, where they form part of a distinct PG biosynthetic complex mediating stalk elongation[45]. In conclusion, genetic, protein localization and in vitro studies have provided solid, but so far only fragmentary information about the regulatory roles of DipM and LdpF, which is insufficient to comprehensively understand the physiological roles of LytM regulators in *C. crescentus*.

In the present study, we characterized the interactome of the *C. crescentus* LytM regulators DipM and LdpF using various in vivo and in vitro approaches. Our results indicate that DipM interacts with four different autolysins (SdpA, SdpB, AmiC, CrbA) and the divisome component FtsN in vivo. We verified these interactions in vitro and revealed that the five interactors compete for DipM and do not produce ternary or higher-order complexes. Moreover, using PG degradation assays, we obtained direct evidence for a stimulatory effect of DipM on the lytic activities of SdpA and AmiC. To improve our mechanistic understanding of DipM, we solved the crystal structure of the LytM domain of DipM, which revealed a conserved groove containing the degenerate catalytic site that was predicted to be the docking site of the different regulatory targets by modeling studies. Mutational analysis confirmed a critical role of this groove in the function of DipM in vivo and in vitro. Moreover, localization and single-molecule tracking studies reveal the existence of a self-reinforcing cycle, in which the DipM-dependent recruitment of SdpA and SdpB to midcell in turn increases the residence time of DipM at this location, most likely leading to a progressive increase in autolytic activity as cell constriction proceeds. Thus, *C. crescentus* DipM coordinates a complex autolysin network whose topology greatly differs from that of previously investigated autolysin systems.

## Results

### DipM mediates a novel interaction network

Previous studies had shown that DipM stimulated the activity of the amidase AmiC in vitro[18,51]. However, while a Δ*dipM* mutant displayed severe pleotropic defects, including filamentation, membrane blebbing and ectopic pole formation, cells depleted of AmiC only showed a cell chaining phenotype[18,51]. Prompted by the finding that DipM was required for the midcell localization of the two SLTs SdpA and SdpB[18], we hypothesized that this LytM regulator could have an activity beyond amidase activation and interact with more autolysins than just AmiC.

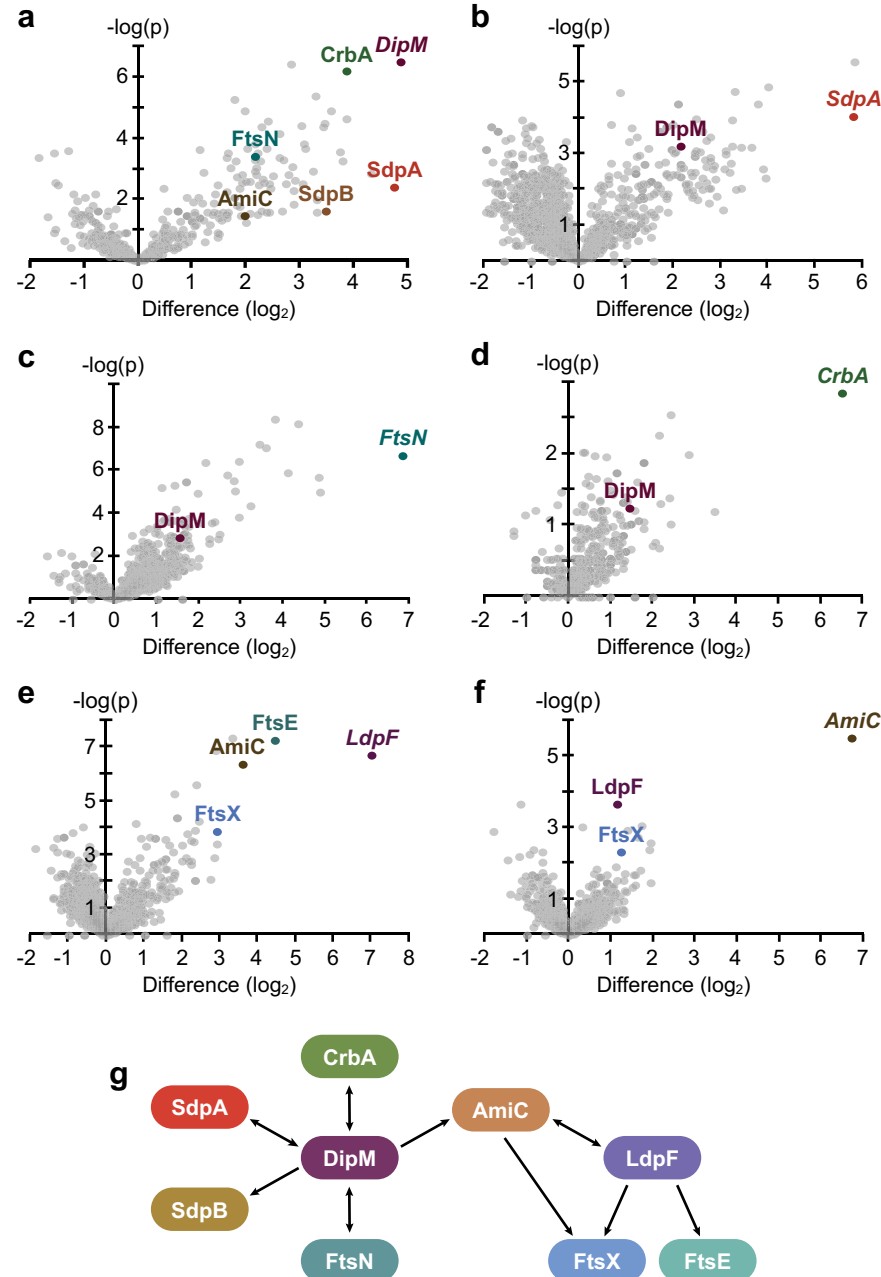

**Fig. 1 | Interactome of DipM and LdpF.** Volcano plots showing the interactors of (**a**) DipM-FLAG (AI021, four replicates), **b** SdpA-FLAG (AI032, three replicates), **c** FtsN-GFP (MT46, four replicates), **d** CrbA-FLAG (AI038, two replicates), **e** LdpF-FLAG (AI036, four replicates) and **f** AmiC-FLAG (AI053, four replicates) as identified by Co-IP analysis. The data points show the $\log_2$ of the average differences in the peptide counts for each hit compared to the control (x-axis) plotted against the $-\log_{10}$ of the p values of these peptide counts (y-axis), calculated using a two-sample t-test. Dots in color represent hits relevant for the purpose of this study, with the names of the corresponding proteins given in the same color next to them. Italic font is used for the bait proteins used in the respective experiment. Details on the data plotted are given in Supplementary Data 1. **g** Schematic representation of the relevant hits and their interactions as identified by Co-IP analysis. Arrows point from the bait proteins to the corresponding enriched prey proteins.

In order to fully elucidate the function of DipM, we determined its interactome in live cells using co-immunoprecipitation (Co-IP) analysis. To this end, we constructed a strain that produced a fully functional DipM variant carrying a C-terminal FLAG affinity peptide (DipM-FLAG) in place of the wild-type protein. After crosslinking with formaldehyde, protein complexes were captured with anti-FLAG affinity beads and subjected to mass spectrometry. Several proteins were specifically enriched in this analysis (Fig. 1a), including PG-degrading enzymes (SdpA, SdpB, AmiC, CrbA), the cell division protein FtsN, the PG synthase PbpX[52] and various so-far uncharacterized proteins. For the present study, we decided to focus on SdpA, SdpB, AmiC, CrbA and

FtsN, which, like DipM, have been shown to be involved in cell division and/or stalk elongation[18,45,53]. To confirm the interactions, we performed reciprocal Co-IP analyses using the identified proteins as baits. We observed a specific enrichment of DipM by FLAG-tagged versions of SdpA (Fig. 1b), FtsN (Fig. 1c) and CrbA (Fig. 1d), suggesting that these proteins indeed associate with each other in vivo. In a complementary analysis, a similar set of experiments was conducted on the second LytM regulator of *C. crescentus*, LdpF[18,51]. The results obtained indicate a mutual interaction of AmiC and LdpF, and they provide evidence for the interaction of both proteins with the FtsEX complex (Fig. 1e, f). Collectively, these findings suggest the existence of two different

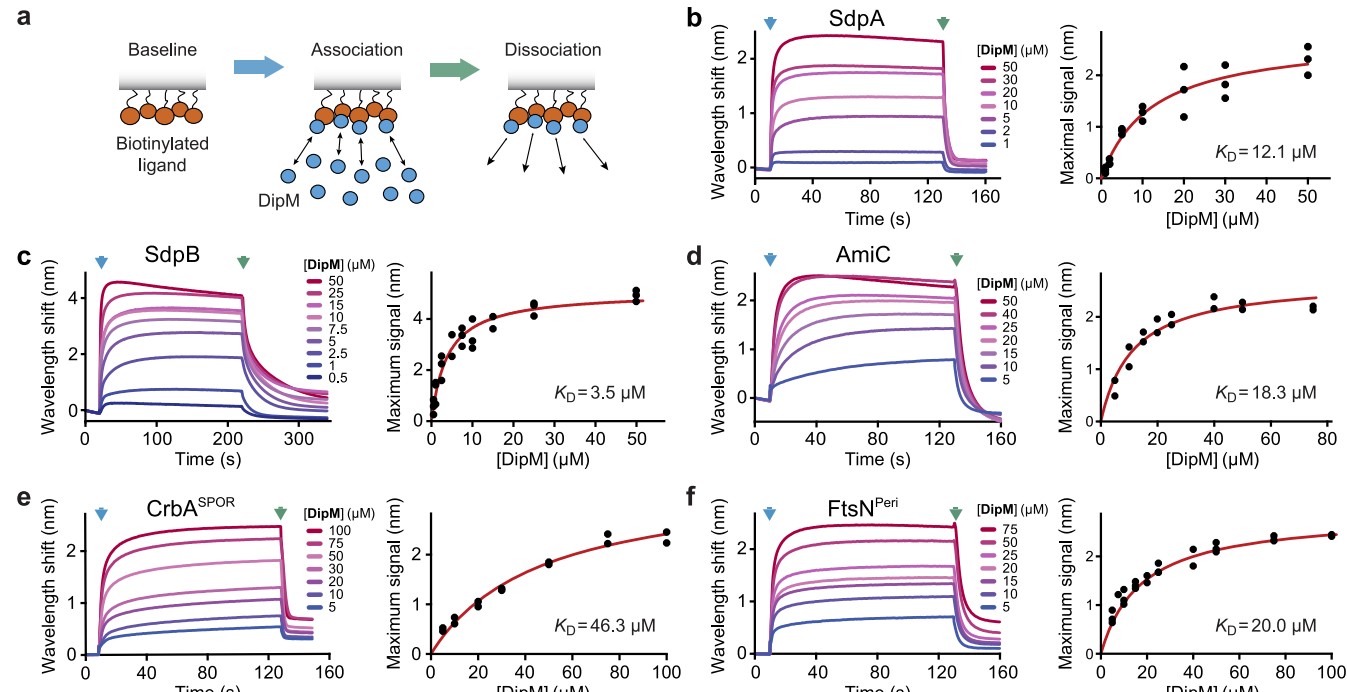

**Fig. 2 | DipM binds SdpA SdpB, FtsN, AmiC and CrbA in vitro. a** Schematic showing the different steps of a bio-layer interferometry (BLI) experiment. After immobilization of the biotinylated ligand, the sensor is washed with buffer to establish a stable baseline. To start the association reaction, the sensor is dipped into a solution of the analyte prepared in the same buffer, which will progressively bind to the immobilized ligand until equilibrium is reached. Finally, the sensor is dipped into analyte-free buffer to monitor the dissociation reaction. **b** BLI analysis of the interaction of DipM with immobilized SdpA. The graph on the left shows bindings curves from a representative BLI experiment, obtained by probing SdpA with the indicated concentrations of DipM. The blue arrow marks the beginning of the association phase, the green arrow the beginning of the dissociation phase. The graph on the right shows the wavelength shifts after equilibration of the binding reactions plotted against the corresponding DipM concentrations in BLI experiments using sensors functionalized with SdpA ($n = 3$ independent experiments). The red line represents the best fit to a one-site binding model. **c**–**f** Analogous to (**b**), but using sensors functionalized with biotinylated SdpB, AmiC, the SPOR domain of CrbA (CrbA^SPOR) and the periplasmic region of FtsN (FtsN^Peri), respectively. Source data are provided as a Source Data file.

regulatory hubs (Fig. 1g). The first one, DipM, interacts with the lytic enzymes SdpA, SdpB, AmiC and CrbA as well as the divisome component FtsN, a bitopic membrane protein serving as a central activator of PG remodeling enzymes at the division site[54]. The second hub, LdpF, by contrast, likely connects AmiC with the FtsEX complex, further supporting the notion that it represents a functional homolog of the gammaproteobacterial EnvC protein.

To determine whether the interactions observed by Co-IP analysis were direct, we purified the proteins or soluble fragments of them and then analyzed their interaction by bio-layer interferometry (BLI). For this purpose, the presumed DipM targets were immobilized on BLI sensors and probed with DipM as an analyte. The results obtained showed that DipM associates with SdpA, SdpB, AmiC, the SPOR domain of CrbA and the periplasmic region of FtsN (Fig. 2). In all cases, the complexes were highly dynamic, with fast association and dissociation kinetics and equilibrium dissociation constants ($K_D$) in the low micromolar range (3–46 μM). By contrast, no interaction was observed with the periplasmic L,D-transpeptidase LdtD[45], which was not obtained as a significant hit in any of the Co-IP analyses, verifying the specificity of the binding reactions (Supplementary Fig. 1).

We also considered the possibility that DipM could act as an interaction platform that is able to simultaneously bind multiple different target proteins. To address this point, we analyzed whether the preincubation of DipM with a second interactor before its addition to the BLI sensors resulted in an increase in the binding signal, indicating the formation of a ternary complex, or rather in a decrease of the signal, which would suggest a competition of the DipM target proteins for the same binding site (Supplementary Fig. 2a). It was not possible to test all combinations due to non-specific binding of some of the proteins to the sensor surface. However, for the protein pairs

analyzed, the signals remained largely unchanged or decreased compared to the reactions with DipM alone (Supplementary Fig. 2b–e). These findings indicate that DipM can only bind one target protein at a time and does not mediate the formation of multi-protein complexes.

## DipM enhances the activities of SdpA and AmiC in vitro

Having confirmed a direct interaction of DipM with various autolysins, we next investigated whether DipM had a regulatory effect on the enzymatic activity of its target proteins. To this end, we performed PG digestion assays with purified SdpA in the absence or presence of DipM (Fig. 3a). When incubated with purified sacculi, SdpA showed basal lytic activity against non-crosslinked and, to a small extent, also crosslinked PG, producing muropeptides that contained 1,6-anhydro-MurNAc moieties (Fig. 3b). This result confirms the bioinformatic prediction that SdpA is an SLT. Importantly, the addition of either DipM or DipM^LytM led to a considerable increase in the activity of SdpA against crosslinked PG, while its activity against non-crosslinked PG was only moderately stimulated under these conditions (Fig. 3b, c). It was conceivable that DipM affected SdpA activity only indirectly by binding to PG and thus improving the accessibility of the cleavage sites. To test this possibility, we also analyzed the effect of DipM on the lytic transglycosylase Slt from *E. coli*, a protein with an enzymatic domain homologous to that of SdpA. However, neither DipM nor DipM^LytM had any stimulatory effect on the activity of Slt (Supplementary Fig. 3), verifying that the increase in SdpA activity observed in the presence of DipM is achieved by a specific interaction between the two proteins. It was not possible to perform similar assays with SdpB, because its activity was below the detection limit in all conditions tested, suggesting that SdpB requires as-yet unknown additional factors for functionality. We also tested the effect of DipM on AmiC, assaying the AmiC-mediated cleavage of

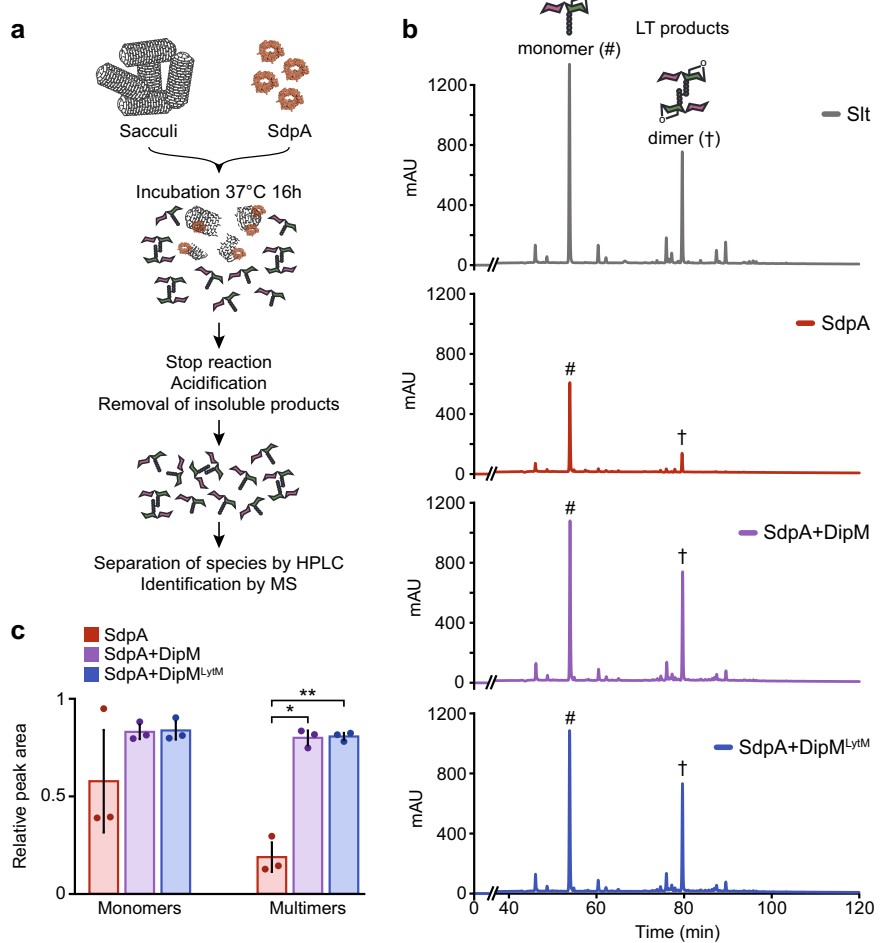

**Fig. 3 | DipM stimulates the lytic transglycosylase activity of SdpA in vitro.**
**a** Overview of the procedure used to assess the lytic activity of SdpA. **b** HPLC
chromatograms showing the muropeptides generated by incubation of pepti-
doglycan sacculi with the indicated protein(s). SdpA and DipM/DipM^LytM were used
at equimolar ratios. Hash signs (#) mark the peaks of monomeric, daggers (†) those
of dimeric 1,6-anhydro-NAM-containing lytic transglycosylase (LT) products. **c** Bar
chart representing the amount of monomeric and dimeric 1,6-anhydro-NAM-

containing products released in reactions containing the indicated proteins. Bars
represent the mean (± SD) of three independent experiments. Data were normal-
ized to those obtained for Slt. Asterisks indicate a significant difference between
the mean values obtained for the indicated conditions (*$p = 0.00052$,
**$p = 0.00036$), as determined by one-way ANOVA. Source data are provided as a
Source Data file.

muropeptides previously released from PG sacculi by muramidase
treatment (Supplementary Fig. 4a). We found that AmiC had a weak
basal activity, which was strongly stimulated by DipM or DipM^LytM
(Supplementary Fig. S4b), supporting and extending previous findings
based on dye-release assays[51]. Collectively, our findings demonstrate
that DipM not only recruits several autolytic enzymes but also stimu-
lates the activities of SdpA and AmiC, making it the first reported multi-
class autolysin activator.

**The LytM domain is critical but not sufficient for DipM function**
Previous work has suggested that the production of the LytM domain
of DipM was sufficient to complement the Δ*dipM* phenotype[48–50].
However, the strains used were constructed by transformation of a
Δ*dipM* mutant, which in our hands is phenotypically highly unstable.
To obtain more reliable information about the contributions of the
different domains (Fig. 4a) to DipM function, we carefully analyzed the
functionality of DipM derivatives lacking either all four LysM or
the C-terminal LytM domain. To this end, wild-type DipM and the
mutant proteins were tagged with the cyan fluorescent protein
sfmTurquoise2^ox55 and produced in a conditional *dipM* mutant, whose
native DipM protein was depleted right before analysis to avoid the
accumulation of suppressor mutations (Supplementary Fig. 5). While

the wild-type fusion was fully functional under these conditions, var-
iants lacking the LysM domains failed to restore normal cell shape and
division, producing phenotypes similar to that of Δ*dipM* cells when
analyzed in double-concentrated growth medium (Fig. 4b, c). The
mutant proteins showed a diffuse localization pattern, with brighter
speckles along the cell periphery that may represent ectopic com-
plexes of DipM with its target proteins. However, although functionally
impaired, they still supported cell growth with appreciable rates
(Fig. 4d). By contrast, a variant lacking the C-terminal LytM domain
(DipMΔ390-609) was no longer able to support growth. In this case,
the cells again formed long filaments that displayed dense arrays of
fluorescent foci along the cell periphery, suggesting that the LytM
domain is required to properly form or position the cell wall features
recognized by the LysM domains of DipM. Collectively, these results
show that the functionality of DipM depends on both its enrichment at
the division site through the PG-binding LysM domains and its inter-
action with regulatory targets through the LytM domain, although only
the latter is essential for cell viability.

**Structural analysis reveals key features of DipM^LytM**
Since the LytM domain is essential for DipM function and sufficient to
stimulate SdpA and AmiC activity, we aimed to obtain more insight

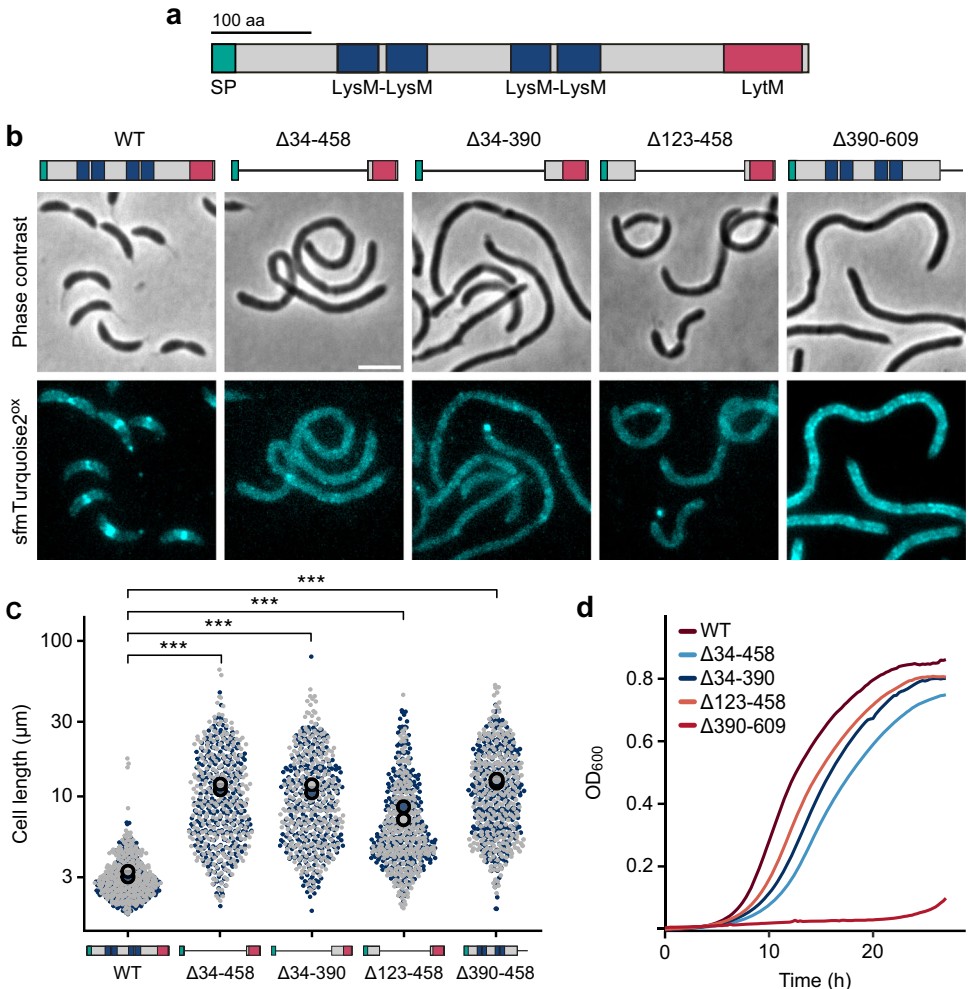

**Fig. 4 | Both the LysM domains and the LytM domain a required for proper DipM function. a** Schematic showing the domain architecture of DipM. The positions of the signal peptide (SP), the four LysM domains and the LytM domain are indicated. Non-structured regions are shown in gray. **b** Functionality and localization patterns of a wild-type DipM-sfmTurquoise2[ox] fusion (MAB512) and variants thereof lacking parts of the N-terminal region containing the LysM domains (MAB501, MAB502, MAB503) or the C-terminal LytM domain (MAB513). Shown are phase contrast and fluorescence images of cells producing the indicated fusion proteins in place of the wild-type protein in 2xPYE medium (scale bar: 3 μm). Schematics depicting the domain organization of the different variants are shown on top. **c** Superplots showing the distribution of cell lengths in the cultures described in (**b**). The data represent the results of two replicates (gray and blue; $n = 300$ each). The big filled circles indicate the mean cell length obtained for each replicate. Asterisks indicate the statistical significance of differences between strains (unpaired two-sided $t$-test; ***$p < 2.22 \times 10^{-16}$). **d** Growth curves of strains producing the indicated DipM-sfmTurquoise2[ox] fusions. Lines represent the mean of two biological replicates. Source data are provided as a Source Data file.

into its mode of action. We therefore purified a DipM fragment (henceforth called DipM[LytM]) comprising the predicted LytM domain (residues 503–609) and part of the adjacent N-terminal non-structured region (residues 459–502) and solved its crystal structure by X-ray diffraction to 2.25 Å resolution (Fig. 5a and Table S1). The crystals contained four independent chains per asymmetric unit, with clear electron densities for the LytM domain and part of the N-terminal extension (Supplementary Fig. 6).

The structure of DipM[LytM] is characterized by a core folding motif that comprises a two-layered β-sandwich composed of a central large β-sheet (β3, β5, β6, β7 and β10) and an adjacent smaller β-sheet (β4, β8 and β9) (Fig. 5a). Beyond this, DipM[LytM] contains two additional β-strands, forming a β-hairpin (β1–β2) that emanates from the core, a feature also observed for LytM of *S. aureus* and other Lysostaphin-like proteins[56] (Fig. 5a). As expected for a non-catalytic LytM regulator, DipM[LytM] shows a degenerate catalytic site containing only one of the three metal-coordinating residues conserved in LytM domains with lytic activity (Fig. 5b). The four independent chains in the crystal exhibit very similar structures, with root-mean-square deviation (rmsd) values

for the $C_\alpha$ atoms ranging from 0.061 to 0.235 Å. The main differences are observed in four loops that display conformational plasticity, including $L_{\beta1–\beta2}$, $L_{\beta5–\alpha3}$, $L_{\beta9–\beta10}$ and $L_C$ (residues 589–597 at the C-terminal end), and in the flexible N-terminal region immediately adjacent to the LytM domain (Supplementary Fig. 6a, b). Unless indicated otherwise, chain C will serve as a reference in this study. Overall, the structure of DipM[LytM] is similar to that of EnvC[LytM] from *E. coli* (PDB: 4BH5), with an rmsd of 0.457 Å for the 80 superimposed $C_\alpha$ atoms (Fig. 5c). However, relevant differences are observed in the aforementioned loops $L_{\beta1–\beta2}$, $L_{\beta5–\alpha3}$, $L_{\beta9–\beta10}$ and especially in loop $L_C$, which is much longer in DipM than in EnvC and other characterized LytM domain-containing proteins (Fig. 5c and Supplementary Fig. 7a). This extension of $L_C$ appears to be conserved only in DipM proteins of the genera *Caulobacter* and *Phenylobacterium* and contains a characteristic KDKA motif at its center (Supplementary Fig. 8). However, complementation experiments show that this longer loop has only a minor role in DipM function (Supplementary Fig. 7b).

Previous work has shown that the residues of EnvC required for amidase activation are all clustered in and around the larger β-sheet of

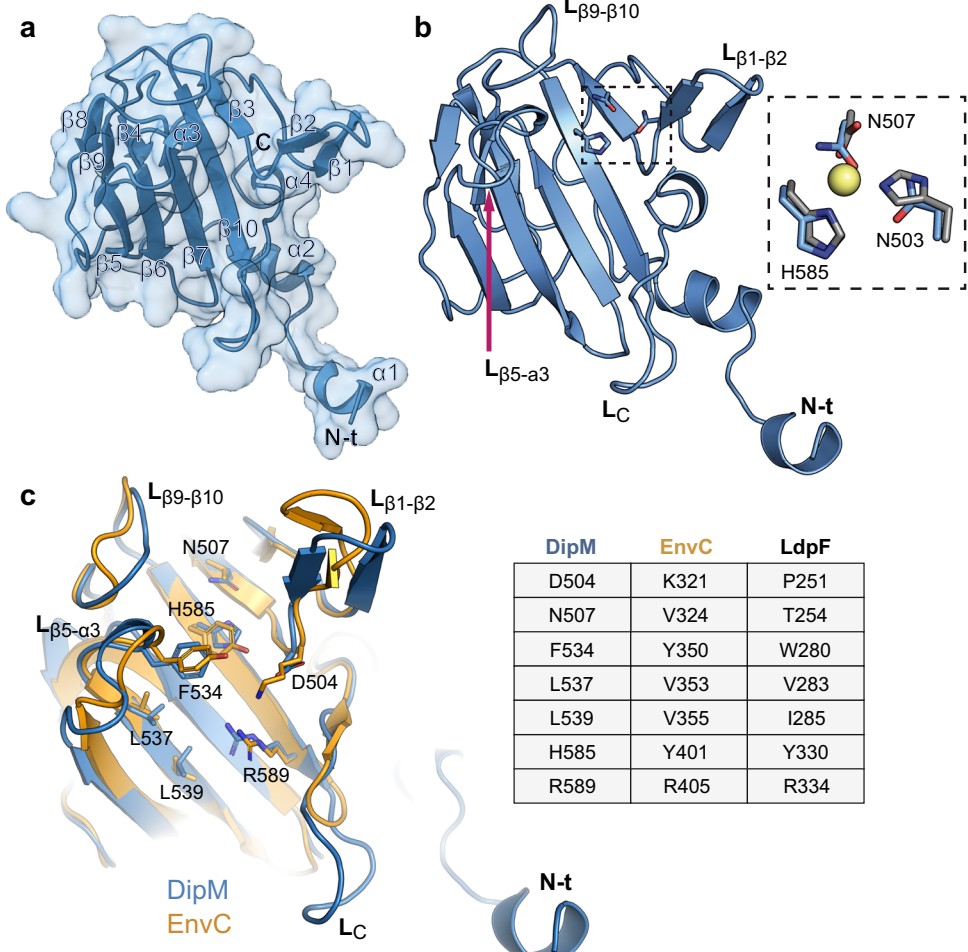

**Fig. 5 | Crystal structure of the LytM domain of DipM. a** Cartoon and surface view of the LytM domain of DipM (DipM^LytM). Secondary structures are numbered according to their position in the polypeptide chain. **b** Cartoon view of DipM^LytM in the same orientation as in (**a**), with some of the loops labeled. The degenerated active site is highlighted by a frame. A magnified view of this region is shown on the right, with the residues of DipM (blue sticks and labeled) superimposed to the metal-binding residues of LytM from *S. aureus* (gray sticks and the catalytic zinc ion

shaped as a yellow sphere; PDB: 1QWY). **c** Structural alignment of DipM^LytM (teal) and *E. coli* EnvC^LytM (orange, PDB: 4BH5)[57] in cartoon view (rmsd 0.457 Å for 80 C_α atoms). Regions showing significant differences are labeled as in (**b**). Key residues of EnvC involved in amidase activation as well as the corresponding residues of DipM are depicted as orange and blue sticks, respectively. A comparison of these residues in the LytM regulators DipM, EnvC and LdpF is given in the table on the right.

the LytM domain core fold[57]. Some of these residues (V353, V355, R405, Y350) are conserved in DipM^LytM (L537, L539, R589, F534) and also in LdpF^LytM (Fig. 5c). Others, by contrast, have changed in DipM and adopted different physico-chemical properties, including D504 (K321 in EnvC), N507 (V324 in EnvC) and H585 (Y401 in EnvC) (Fig. 5c), which may have implications for AmiC recognition. The positive electrostatic surface potential in this region, which was shown to be essential for amidase activation by EnvC[57], is also conserved in DipM, with loop L_c providing even more positive charges to the binding pocket (Supplementary Fig. 9).

The N-terminal part of the DipM fragment crystallized in this study, ranging from its start to loop β1–β2 (aa 459–502), is not part of the LytM domain as annotated by the Pfam database[58], because the underlying hidden Markov model does not include this region (Supplementary Fig. 10a). However, helix α2 interacts with the C-terminal part of the LytM domain, and the adjacent non-structured region makes close contacts with the LytM core domain. Truncation of this N-terminal region led to a significant reduction in the steady-state levels of DipM^LytM (Supplementary Fig. 10b), indicating that this part of the protein may be required for efficient folding or stabilization of the LytM domain. Notably, crystal structures of other LytM domain-

containing proteins display comparable non-structured regions with strikingly similar conformations adjacent to the predicted LytM domain (Supplementary Fig. 10c). This structural element thus appears to be an integral part of the LytM domain, although its high sequence variability prevents its recognition by a hidden Markov model, based on protein sequence alignments.

**The LytM domain acts as the target-binding site of DipM**
The high structural similarity of the LytM domains of DipM and EnvC (EnvC^LytM) prompted us to explore whether partner recognition by DipM could follow the same principles as that by EnvC. In a recent structure of full-length EnvC in complex with the extra-cellular domain of FtsX (FtsX^ED)[59], residues involved in amidase activation by EnvC^LytM were found to be occluded by a "restraining arm" formed by an α-helical region adjacent to the LytM domain, suggesting a self-inhibition mechanism[59] (Fig. 6a and Supplementary Fig. 11a, b). The determinants in the binding groove of EnvC^LytM that accommodates the restraining arm include residues E317, K321 and Q315 in the β-hairpin motif as well as R340, E357 and R405 in the large β-sheet (Fig. 6b), matching some of the residues (K321 and R405) shown to be involved in amidase activation[57]. Interestingly, DipM^LytM shows a conspicuous groove

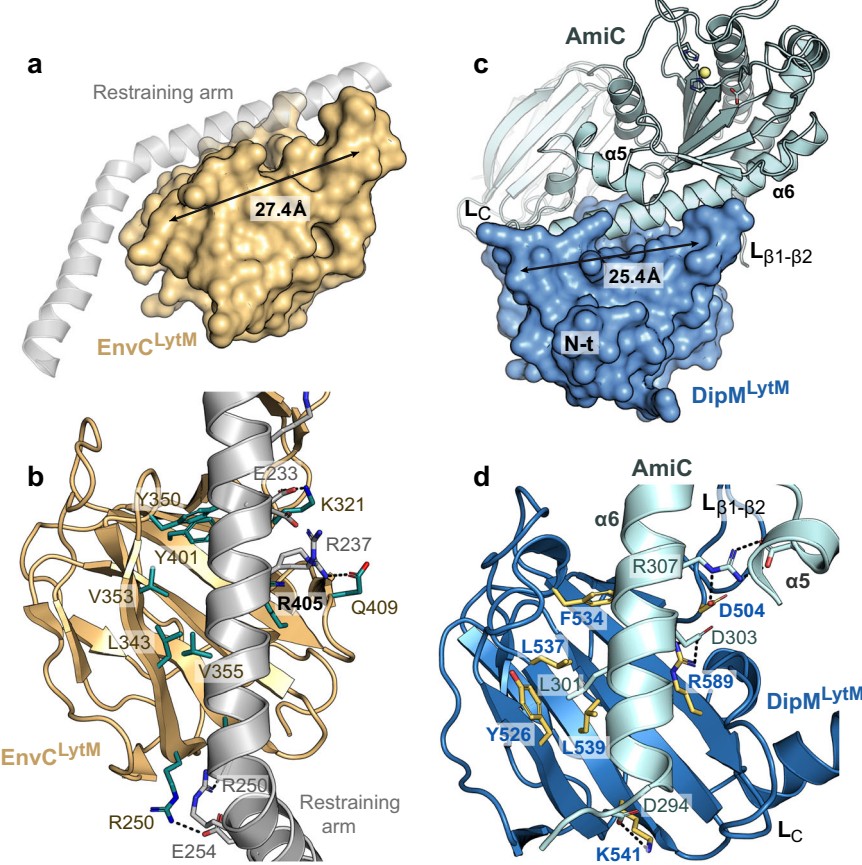

**Fig. 6 | Structural model of the DipM^LytM^-AmiC complex. a** LytM domain of EnvC (EnvC^LytM^) associated with the self-inhibitory restraining arm (gray transparent cartoon; PDB: 6TPI)[59]. The length of the binding groove is indicated. **b** Detailed view of the interface between EnvC^LytM^ and the restraining arm. Residues involved in the interaction are highlighted in teal and shown in stick representation. Hydrogen bonds are displayed as dashed lines. **c** Model of the DipM^LytM^-AmiC complex,

generated by AlphaFold-Multimer[60]. The crystal structure of DipM^LytM^ is shown in surface representation (dark blue), the modeled structure of AmiC from *C. crescentus* in cartoon view (light blue). The length of the predicted binding groove is indicated. **d** Detailed view of the predicted interface between DipM^LytM^ and helices α5/α6 of AmiC. Residues involved in the interaction are shown in orange in stick representation. Relevant hydrogen bonds are shown as dashed lines.

(9.5 Å × 25.4 Å) around the degenerate catalytic site of the protein whose dimensions are similar to those of the binding groove found in EnvC^LytM^ (10 Å × 27.4 Å) (Fig. 5c and Supplementary Fig. 11b, d). To obtain more insight into the potential mode of action of DipM, we used AlphaFold-Multimer[60] to generate a structural model of DipM^LytM^, which agreed very well with the crystal structure solved in this study (Supplementary Fig. 12a). We then went on to predict the structure of the DipM^LytM^-target protein complexes. In the five top-ranking models of the DipM^LytM^-AmiC complex, which were highly similar to each other (Supplementary Fig. 12b, c), a long α-helix (α6) from AmiC was inserted into the groove of DipM^LytM^, recapitulating the arrangement of the restraining arm in the binding groove of EnvC^LytM^ (Fig. 6c, d). Similarly, the interactions with SdpA, SdpB, CrbA and FtsN were predicted to involve an association of these proteins with the groove of DipM^LytM^ and the surrounding loops L_β1-β2_, L_β5–α3_ and L_C_ (Supplementary Fig. 13). These results suggested that the degenerate catalytic site of DipM^LytM^ could have a central role in the interaction of DipM with its regulatory targets.

To further clarify the location of the target-binding site, we mutated conserved hydrophobic and charged residues in the predicted target-binding groove of DipM, including R589, L537 and L539 (Fig. 6d), and then analyzed the functionality of the mutant proteins in an in vivo complementation assay (Supplementary Fig. 14). Notably, all three exchanges strongly impaired DipM function and gave rise to a phenotype similar to that of a Δ*dipM* mutant[48–50] (Fig. 7a, b). However, unlike a truncated variant lacking the LytM domain (compare Fig. 4),

the mutant proteins still supported cell growth, albeit at reduced rates (Fig. 7c), suggesting some degree of redundancy in the interaction determinants. Moreover, they still formed band-like patterns within the cells that reflected their association with sites of ongoing or abortive cell division (Fig. 7a), consistent with the notion that DipM localization is predominantly mediated by the LysM domains. Together, these findings underscore the central role of the conserved groove in the regulatory activity of DipM. To determine whether the functional defects observed were indeed caused by a loss of the interaction with autolysins, we purified two of the non-functional DipM variants (L539S, R589A) as well as a variant with two substitutions (L537S L539S) and analyzed their binding to SdpA and AmiC as representative regulatory targets using BLI. Importantly, all three mutant variants showed a severely reduced target-binding affinity (Fig. 7d), supporting the notion that the groove containing the defective catalytic site of DipM serves as a docking site for at least some of the interaction partners.

## DipM and SdpA/B form a regulatory feedback loop

Next, we aimed to clarify the contributions of the four PG-binding LysM domains and the C-terminal LytM domain to the localization dynamics of DipM. To this end, the full-length protein, a C-terminally truncated variant lacking the LytM domain (DipM^ΔLytM^) and the LytM alone (DipM^LytM^) were fused to sfmTurquoise2^ox^ and produced in the wild-type background to ensure normal cell morphology and division (Supplementary Fig. 15a). Subsequently, we followed the motion of

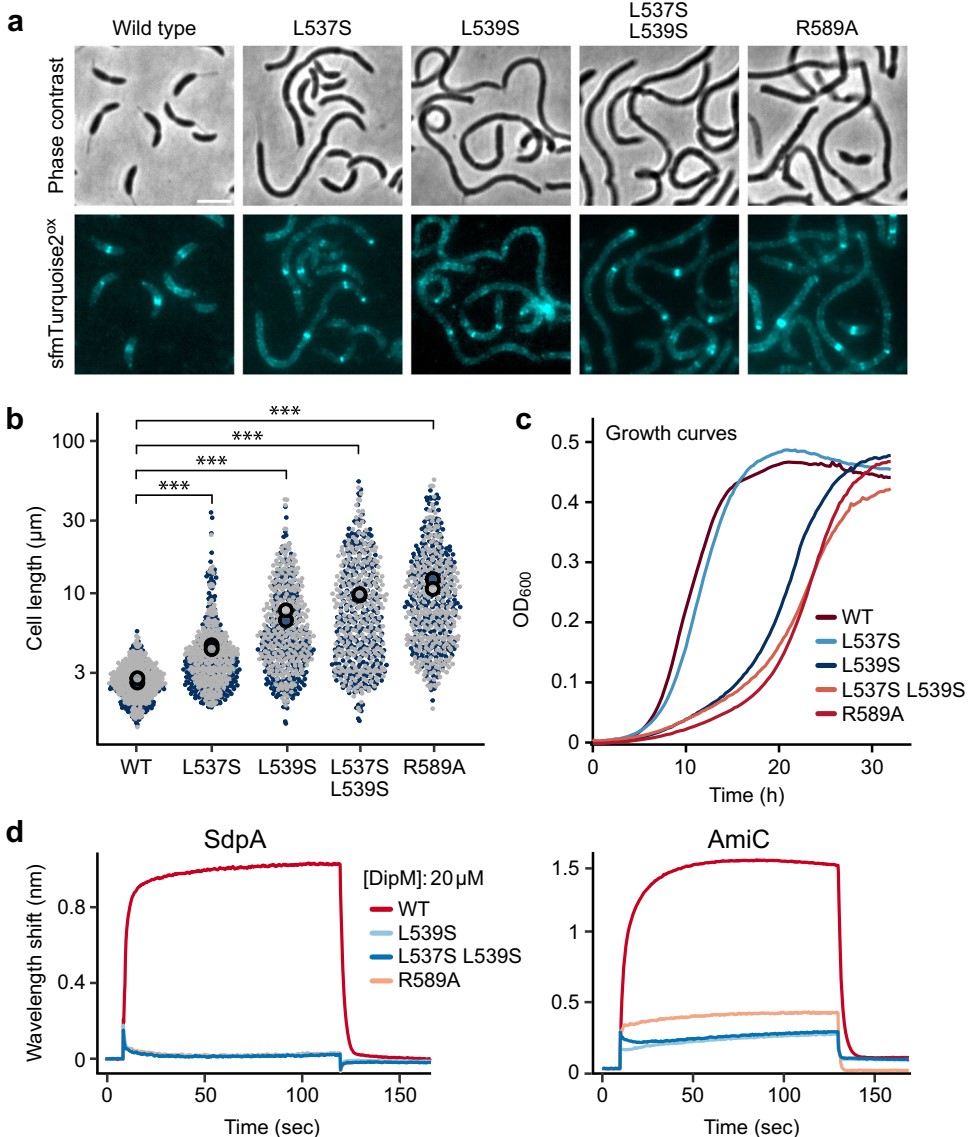

**Fig. 7 | Amino acid exchanges in the putative binding groove of the LytM domain abolish DipM function. a** Functionality and localization patterns of DipM-sfmTurquoise2$^{ox}$ variants with amino acid exchanges in the LytM domain. Shown are phase contrast and fluorescence images of cells producing the indicated fusion proteins in place of the wild-type protein in PYE medium (MAB512, MAB504, MAB505, MAB506, MAB510). Scale bar: 3 µm. **b** Superplots showing the distribution of cell lengths in the cultures of strains producing the indicated DipM-sfmTurquoise2$^{ox}$ fusions. The data represent the results of two replicates (gray and blue; $n = 300$ each). The big filled circles indicate the mean cell length obtained for each replicate. Asterisks indicate the statistical significance of differences between strains (unpaired two-sided $t$-test; ***$p < 2.22 \times 10^{-16}$). **c** Growth curves of strains producing the indicated DipM-sfmTurquoise2$^{ox}$ fusions. Lines represent the mean of two biological replicates. **d** BLI analysis of the interaction of mutant DipM variants with SdpA and AmiC. Biotinylated SdpA (left) or AmiC (right) was immobilized on BLI sensors and probed with the indicated DipM variants (20 µM). The graphs show the results of representative experiments ($n = 3$ independent replicates). Source data are provided as a Source Data file.

single molecules in synchronized live cells harvested shortly after the onset of cell constriction, a cell cycle stage when DipM is usually localized at the cell division site[48–50]. An analysis of the single-molecule tracks obtained revealed that the overall mobility of DipM was very low, with most molecules showing slow, confined motion in the central region of the cell (Fig. 8a). In each of the cells imaged, a smaller fraction of slow-moving molecules was also observed at one of the cell poles. Our experimental setting did not allow us to discriminate between the old and new cell pole. However, DipM is known to localize to the old pole early in the cell cycle to support stalk formation[45], suggesting that these signals reflect the old-pole-associated DipM population. The distribution of step sizes in the single-particle tracks indicates the existence of two distinct diffusion regimes, with a slow-moving (static) and a fast-moving (mobile) population (Supplementary Fig. 16). For

full-length DipM, ~75% of the molecules were static, consistent with previous fluorescence-recovery-after-photobleaching data[50], whereas the remaining 25% were mobile, with an ~10-fold higher diffusion rate (Fig. 8b, c). The two truncated DipM variants were more mobile than the full-length protein, but they still showed a considerable number of confined tracks at the division site (Fig. 8a). For DipM$^{\Delta LytM}$, the fraction of static molecules was only slightly decreased (~60%) (Fig. 8b, c), reflecting the predominant role of the PG-binding LysM domains in the localization of DipM at the cell division site and the stalk base. Importantly, however, DipM$^{LytM}$ also showed a large fraction (~54%) of static molecules. This finding is consistent with the observation that the regulatory LytM domain closely interacts with components of the cell division apparatus and explains why DipM was still partially functional in the absence of the LysM domains (Fig. 4).

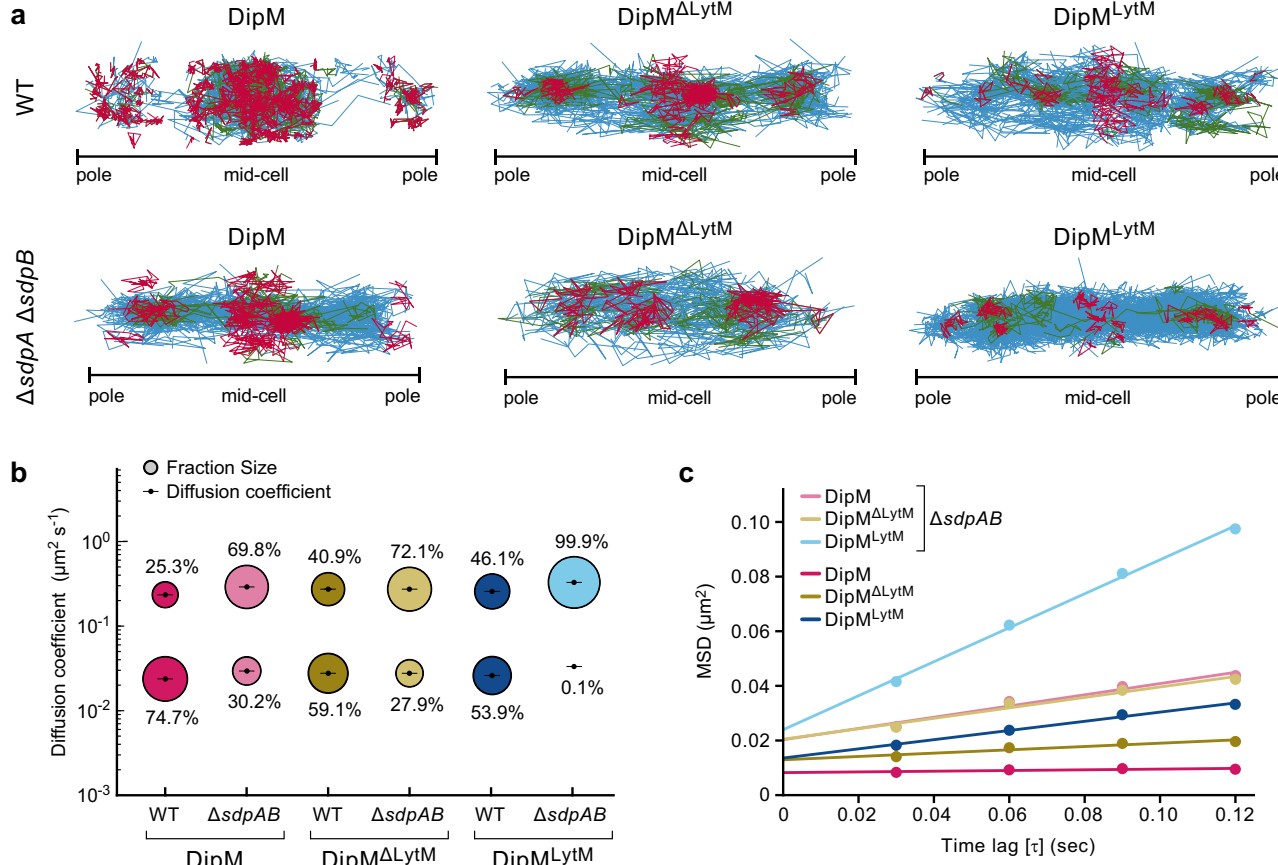

**Fig. 8 | SdpA and SdpB affect the single-molecule behavior of DipM.**
**a** Confinement maps showing single-molecule tracks of DipM, DipM$_{1\text{-}390}$ (DipM$^{\Delta LytM}$) or DipM$_{\Delta 35\text{-}458}$ (DipM$^{LytM}$) fused to sfmTurquoise2$^{ox}$ in the wild-type (AI063, AI098, AII112) and $\Delta sdpA$ (AII121, AII122, AII126) backgrounds. Cells producing the indicated fluorescent protein fusions were synchronized and allowed to grow for another 75 min prior to imaging. Confined tracks are shown in red, non-confined tracks in blue, and tracks showing both behaviors in green. Note that, due to technical limitations, it was not possible to determine the orientation of the cells and, thus, distinguish between the stalked (old) and non-stalked (new) pole during imaging. The plots show subsets of the tracks obtained, taken from representative cells. The total numbers of cells and tracks analyzed per strain are given in Supplementary Table 2. **b** Bubble plots showing the proportions of static and mobile molecules for the indicated DipM-sfmTurquoise2$^{ox}$ fusions in the wild-type and $\Delta sdpAB$ backgrounds. **c** Mean-squared-displacement (MSD) analysis of the indicated DipM-sfmTurquoise2$^{ox}$ fusions in the wild-type and $\Delta sdpAB$ backgrounds. The chart shows the MSD values for the first four time-lags ($\tau$) from all tracks containing more than four points and linear fits of the data for each of the indicated conditions. The graphs in (**b**) and (**c**) represent the results of two biological replicates. The underlying data are available from Figshare at https://doi.org/10.6084/m9.figshare.23140343.v1.

Previous work has shown that the SLTs SdpA and SdpB are no longer recruited to the midcell region in a $\Delta dipM$ mutant[18]. A similar effect was observed in cells producing a non-functional variant of DipM lacking the PG-binding LysM domains in place of the wild-type protein (Supplementary Fig. 17). Thus, DipM acts as a localization determinant for SdpA and SdpB, ensuring their activation at midcell at the onset of the cell division process. Since PG remodeling may be an important trigger for the accumulation of DipM at the division site, is was conceivable that SdpA/B activity could in turn have a stimulatory effect on DipM recruitment. To test this hypothesis, we re-analyzed the diffusion behavior of full-length DipM, DipM$^{\Delta LytM}$ and DipM$^{LytM}$ in a $\Delta sdpAB$ background. Importantly, the mobility of DipM and its variants increased considerably in cells lacking the two lytic transglycosylases, accompanied by a decrease in the fraction of confined tracks at the cell center or poles (Fig. 8a). Under these conditions, the proportion of static full-length DipM and DipM$^{\Delta LytM}$ molecules decreased to ~30% (Fig. 8b, c). By contrast, confined motion became essentially unde-tectable for DipM$^{LytM}$, indicating that the interaction of the LytM domain with the division apparatus is mediated by either or both of the two DipM target proteins. Collectively, these results reveal the exis-tence of a self-reinforcing cycle in which the DipM presence stimulates the accumulation of SdpA and SdpB at the cell division site, which in turn promotes the stabilization of DipM in the midcell region. This process may be driven both by physical interactions between the two SLTs and the regulatory LytM domain of DipM as well as by SLT-dependent changes in the structure of the constricting PG layer that are later recognized by its LysM domains.

## Discussion

LytM regulators are found throughout the phylum Proteobacteria and also in some cyanobacteria. Some of their gammaproteobacterial representatives, such as EnvC, contain coiled-coil regions and mediate the interaction between the FtsEX complex and amidases[8,10,34–36]. The LytM regulator LdpF from *C. crescentus* resembles these proteins in its domain architecture, and our Co-IP data indicate that it may indeed serve to physically connect FtsEX with the amidase homolog AmiC (Fig. 1e, f). However, it has been shown that LdpF does not play a role in AmiC activation[51], although it is required for its recruitment to the midcell region[18]. Instead, another LytM regulator, DipM, has been adopted as an additional regulator to stimulate AmiC activity in the *C. crescentus* system[51] (Supplementary Fig. 4). It is possible that LdpF served as an amidase activator in the past but lost this activity during the course of evolution to allow for more complexity in the control of amidase activity. This hypothesis is in agreement with the idea, born

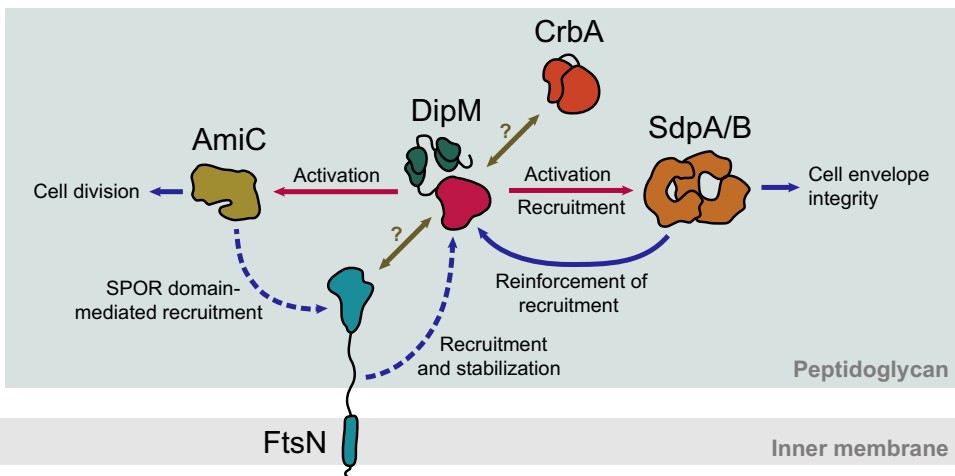

**Fig. 9 | DipM is involved in two feedback loops promoting peptidoglycan remodeling during cell division.** Model for the proposed function of DipM in linking the accumulation and/or activity of the PG biosynthesis regulator FtsN, the amidase AmiC, and the lytic transglycosylases SdpA and SdpB. The effects that the different proteins exert on each other are indicated by arrows (red: enzyme activation, blue: recruitment and other stimulatory effects, olive: unknown). Dashed lines indicate indirect effects induced by the activities of the respective proteins.

from studies in *E. coli*, that amidases and their activators are recruited independently of each other, creating a fail-safe system that minimizes aberrant PG degradation during cell division[61]. It remains to be clarified whether this concept also applies to other alphaproteobacteria and how tight amidase regulation is achieved in species such as *H. neptunium*, which possess only a single catalytically inactive LytM regulator[62].

Unlike EnvC and LdpF, various gammaproteobacterial LytM regulators, such as NlpD and ActS from *E. coli*, contain PG-binding LysM domains instead of coiled-coil regions and mediate amidase activation in an FtsEX-independent manner[34,37,38,63]. *C. crescentus* DipM resembles these proteins, although its LysM domains are organized into two tandems, an arrangement not found in gammaproteobacteria or other alphaproteobacteria. Given its role in amidase activation and its apparent lack of interaction with FtsEX (Fig. 1a, f), DipM could be considered an NlpD homolog. However, the role of NlpD in *E. coli* is likely limited to connecting AmiC to the Tol/Pal system[37], whereas DipM interacts with FtsN and at least four different autolysins, thus exhibiting an unprecedented regulatory complexity (Fig. 1g). We show that DipM not only stimulates the lytic activity of AmiC (Supplementary Fig. 4) but also that of SdpA (Fig. 3b, c). It thus appears that, at least in the alphaproteobacteria, LytM regulators can be multi-class autolysin activators, controlling both amidases and SLTs. While self-regulation has been observed for lytic transglycosylases of family 1E[64], this is, to our knowledge, the first reported case of lytic transglycosylase activation through protein-protein interaction. Notably, a recent study in *E. coli* has also reported a novel lipoprotein, called NlpI, that can interact with several endopeptidases and connect them to other components of the peptidoglycan biosynthetic machinery[65]. Thus, factors that orchestrate the activities of multiple autolysins might be widespread in bacteria.

The ability of DipM^LytM to activate two different classes of enzymes raises the question of whether the interaction with its regulatory targets is mediated by the same or different interfaces. The results of our BLI-based competition assays indicate that the binding interfaces could at least overlap (Supplementary Fig. 2). Consistent with this observation, modeling studies suggest that the groove around the degenerate catalytic site of DipM might act as a common binding interface for all targets proteins (Fig. 6c, d and Supplementary Fig. 13). Amino acid substitutions in this region indeed affect DipM function in vivo and its ability to bind SdpA or AmiC in vitro (Fig. 7), underscoring the central role of the groove in the regulatory activity of

DipM. The precise mechanism underlying the stimulation of autolysin activity by DipM remains to be determined. In the predicted model of the DipM-AmiC complex, DipM closely associates with helix α6 of AmiC and also contacts other regions of the protein. These interactions may trigger a conformational change in AmiC that increases its ability to bind and/or hydrolyze its substrate. Similar effects may be induced in the case of other DipM targets such as SdpA, which also interact with the groove of DipM and its surroundings. However, more work is required to verify the predicted complexes, determine the mechanisms of activation and clarify their conservation among species.

Importantly, SdpA and SdpB have a profound effect on the localization dynamics of DipM and strongly stimulate its association with the cell division site and the old (stalked) cell pole (Fig. 8). This stabilizing effect is also observed for truncated DipM variants comprising only the two LysM tandems or the LytM domain, suggesting that it is based on two distinct but synergistic mechanisms. On the one hand, the activation of the two SLTs at midcell may induce changes in the composition or architecture of the PG layer that are recognized by the LysM domains, thus promoting the association of the N-terminal region of DipM with the septal cell wall. On the other hand, the bivalent interaction of SdpA and SdpB with both peptidoglycan as their enzymatic target and the LytM domain of DipM may help to link the C-terminal region of DipM to the cell wall. Notably, the accumulation of DipM at the division site is required to stimulate the recruitment of the two SLTs in the midcell region (Supplementary Fig. 17), likely through direct protein-protein interactions as well as its regulatory effect on PG remodeling. Thus, while DipM mediates the localization of SdpA and SdpB, the two enzymes in turn stabilize DipM in the midcell region, creating a self-reinforcing cycle that leads to a progressive increase in lytic transglycosylase activity at the division site (Fig. 9).

In addition to the SLT-dependent feedback loop, DipM may also be part of a second positive feedback loop involving AmiC and FtsN. DipM stimulates the activity of AmiC (Supplementary Fig. 4) and thus promotes the generation of naked PG, which is recognized by SPOR domain-containing proteins such as FtsN[66-68]. This interplay between amidases and the SPOR domain is thought to be the main driving force for the accumulation of FtsN at the division site, which in turn activates late divisome components and, thus, cell constriction[21,69]. Conversely, previous work has shown that DipM becomes completely dispersed within the cell upon FtsN depletion, while it still forms patches in the

absence of FtsZ[48], indicating that FtsN is required, directly or indirectly, for DipM localization. These interactions give rise to a self-reinforcing cycle that promotes the progressive accumulation of DipM and FtsN at the division site, in a process that depends on amidase activity and, in turn, gradually increases AmiC activity in the midcell region as cell constriction proceeds (Fig. 9). It still remains to be determined how the direct interaction between DipM and FtsN identified in this (Fig. 2f) and previous[48–50] work contributes to this cycle. Notably, DipM still accumulates normally at the cell division site in strains whose native FtsN protein was replaced by mutant variants that lack the SPOR domain and thus no longer condense at midcell during cell constriction[53]. Thus, the requirement of FtsN for DipM localization may be largely based on FtsN-dependent PG remodeling at the division site[24,54,70], which may induce specific modifications in the constricting PG layer that are recognized by the LysM domains of DipM. However, we observed that DipM foci appeared to be lost prematurely during late stages of the division process in these mutant backgrounds (Supplementary Fig. 18). Moreover, cells often formed chains, as typically observed upon AmiC depletion[18], suggesting a defect in the final steps of daughter cell separation. It is therefore conceivable that direct interactions between FtsN and DipM retain DipM at the division site toward the end of the cell cycle to promote cytokinesis. Moreover, they may help to tune the activities of these two proteins, in a manner dependent on their accumulation levels at midcell.

Together, the establishment of positive feedback loops enables DipM to connect the recruitment and/or activation of autolysins to late-divisome assembly and cell division. Notably, previous work has implicated SdpA and SdpB in cell envelope integrity[18], and studies of their *E. coli* homolog Slt70 suggest a role for SLTs in the degradation of excess, non-crosslinked PG[71]. In support of the latter, we have observed a higher activity of SdpA against non-crosslinked than crosslinked PG in vitro. Considering that *C. crescentus* mostly grows in length through divisome-dependent medial PG incorporation[44], DipM may support this mode of growth by promoting "PG quality control" through SdpA and SdpB.

Collectively, the finding that DipM acts as a multi-class autolysin activator significantly furthers our understanding of the functions that catalytically inactive LytM regulators can adopt. Although DipM may be a specific invention of the alphaproteobacterial lineage, it is likely that other bacteria have evolved similar systems to coordinate multiple autolysins with a single regulatory protein. The characterization of such multi-enzyme regulators, whose malfunction compromises several cell wall-related processes at once, will not only improve our understanding of PG remodeling but also reveal promising new targets for the development of antibacterial drugs.

## Methods

### Growth conditions

All *C. crescentus* strains used in this study are derivatives of the synchronizable wild-type strain NA1000 (CB15N)[72]. *C. crescentus* cells were grown aerobically at 28 °C in peptone-yeast extract (PYE) medium[73] or double-concentrated PYE (2xPYE) medium, supplemented when necessary with antibiotics at the following concentrations ($\mu$g ml$^{-1}$; liquid/solid medium): kanamycin (5/25), gentamicin (0.5/5). To induce the *xylX*[74] or *vanA*[75] promoters, media were supplemented with 0.3% D-xylose or 0.5 mM sodium vanillate, respectively. For microscopic analysis, cells were grown to exponential phase and induced for 2–3 h prior to imaging, unless indicated otherwise. *C. crescentus* swarmer cells were isolated by density gradient centrifugation using Percoll[76]. Plasmids were introduced by electroporation[77] or conjugation, as described previously for *H. neptunium*[78].

*E. coli* TOP10 (Thermo Fisher Scientific) was used for general cloning purposes, *E. coli* WM3064 (W. Metcalf, unpublished) as donor strain for conjugation and *E. coli* Rosetta(DE3)pLysS (Novagen) as host for recombinant protein production. *E. coli* derivatives were grown aerobically at 37 °C in Luria-Bertani medium (LB), supplemented when required with antibiotics at the following concentrations ($\mu$g ml$^{-1}$; liquid/solid medium): gentamicin (15/20), kanamycin (30/50), ampicillin (200/200), chloramphenicol (20/30). Cultures of *E. coli* WM3064 were supplemented with 300 $\mu$M 2,6-diaminopimelic acid (DAP).

### Plasmid and strain construction

The sequences of the genes used in this study were based on the genome sequence of *C. crescentus* NA1000[79]. Details on the strains, plasmids and oligonucleotides are given in Supplementary Tables 3–5. Plasmids were designed with the help of SnapGene 3.3.4 (GLS Biotech, USA). All constructs were verified by DNA sequencing. Plasmids bearing inducible genes were integrated ectopically at the chromosomal *xylX* (P$_{xyl}$) or *vanA* (P$_{van}$) loci of *C. crescentus* by single-homologous recombination[75]. The truncation or in-frame deletion of genes was carried out using a two-step procedure based on the counterselectable *sacB* marker[80]. Proper plasmid integration or gene deletion were verified by colony PCR. The correct truncation of endogenous genes was further confirmed by sequencing of the target loci.

### Complementation assays and growth curves

For complementation experiments, strains expressing *dipM* from the P$_{xyl}$ promoter and mutant variants of *dipM* from the P$_{van}$ promoter were inoculated from cryo-stocks into liquid PYE medium supplemented with 0.3% xylose and grown overnight. The next day, cells were collected by centrifugation, washed twice in PYE or 2xPYE medium without any supplements and then resuspended in PYE or 2xPYE medium containing 0.5 mM sodium vanillate to an OD$_{600}$ of $2 \times 10^{-3}$. After ~18 h of growth (mid-exponential phase), samples were taken for imaging and immunoblot analysis. Subsequently, the cultures were diluted to an OD$_{600}$ of 0.005, transferred to 24-well polystyrene microtiter plates (Becton Dickinson Labware, USA) and monitored at 32 °C under double-orbital shaking in an EPOCH 2 microplate reader (BioTek, USA) to obtain growth curves.

### Immunoblot analysis

Immunoblot analysis was conducted as described previously[80], using a polyclonal anti-GFP antibody (Sigma-Aldrich, Germany; Cat. #: G1544; RRID: AB_439690) at a 1:10,000 dilution. Goat anti-rabbit immunoglobulin G conjugated with horseradish peroxidase (PerkinElmer, USA) was used as secondary antibody. Immunocomplexes were detected with the Western Lightning Plus-ECL chemiluminescence reagent (PerkinElmer, USA). The signals were recorded with a Chemi-Doc MP imaging system (BioRad, Germany) and analyzed using the Image Lab 5.0 software (BioRad, Germany).

### Widefield fluorescence imaging

Cells were grown to exponential phase, spotted on 1% agarose pads, and imaged with an Axio Observer.Z1 microscope (Zeiss, Germany) equipped with a Plan Apochromat ×100/1.4 Oil Ph3 phase contrast objective, an ET-CFP filter set (Chroma, USA) and a pco.edge 4.2 sCMOS camera (PCO, Germany). Images were recorded with Visi-View 4.0.0.5 (Visitron Systems, Germany) and processed with Fiji 1.53t[81], Adobe Photoshop CS6 and Adobe Illustrator CS6 (Adobe Systems, USA). Cell length measurements were performed with Fiji 1.53t. To generate demographs, fluorescence intensity profiles were measured with Fiji 1.53t and processed in R 3.3.0+[82], using the Cell Profiles script (http://github.com/ta-cameron/Cell-Profiles)[83]. SuperPlots[84], generated with ggplot2[85], were used to visualize cell length distributions and to evaluate the statistical significance of differences between multiple distributions.

### Single-particle tracking and diffusion analysis

Cells of the indicated strains were cultivated overnight in PYE medium, transferred into fresh medium and grown to exponential phase prior

to synchronization. The isolated swarmer cells were transferred into pre-warmed M2G liquid medium supplemented with 0.3% D-xylose and grown for 90 min before imaging by slimfield microscopy. In this approach, the back aperture of the objective is underfilled by reduction of the width of the laser beam, generating an area of about 10 μm in diameter with high light intensity that allows the visualization of single fluorescent protein molecules at very high acquisition rates. The single-molecule level was reached by bleaching most molecules in the cell for 100 to 1000 frames, followed by tracking of the remaining and newly synthesized molecules for ~3000 frames. Images were taken at 30 ms intervals using an Olympus IX-71 microscope equipped with a UAPON ×100/NA 1.49 TIRF objective, a back-illuminated electron-multiplying charge-coupled device (EMCCD) iXon Ultra camera (Andor Solis, USA) in stream acquisition mode, and a LuxX 457-100 (457 nm, 100 mW) light-emitting diode laser (Omicron-Laserage Laserprodukte GmbH, Germany) as an excitation light source. The laser beam was focused onto the back focal plane and operated during image acquisition with up to 2 mW (60 W/cm² at the image plane). Andor Solis 4.21 software was used for camera control and stream acquisition. Prior to analysis, frames recorded before reaching the single-molecule level were removed from the streams, using photobleaching curves as a reference. Subsequently, the streams were cropped to an equal length of 2000 frames and the proper pixel size (100 nm) and time increment were set in the imaging metadata using Fiji 1.53t[81]. Single particles were tracked with u-track 2.2[86]. Trajectories were only considered for further statistical analysis if they had a length of at least five steps. Data analysis was performed using SMTracker 2.0[87]. The diffusive behavior of the proteins investigated was analyzed in two ways. Mean-squared-displacement (MSD)-versus-time-lag curves were calculated to provide an estimate of the diffusion coefficient and clarify the kind of motion exhibited (e.g., diffusive, subdiffusive or directed). In addition, we determined the frame-to-frame displacement of all molecules in x and the y direction and fitted the resulting distributions to a two-population Gaussian mixture model to determine the proportions of mobile and static molecules in each condition[87].

## Protein purification

**Full-length DipM.** To purify wild-type MAS-DipM(26-609)-His₆ or mutant variants of this protein, cells of *E. coli* AM201 (WT)[49], MAB493 (L539S), MAB500 (L537S L539S) or MAB516 (R589A) were grown in LB medium at 37 °C. At an OD₆₀₀ of 1, protein overproduction was induced by the addition of 1 mM isopropyl-β-D-thiogalactopyranoside (IPTG) and cultivation was continued for 3 h. The cells were harvested by centrifugation, washed with buffer BZ3 (50 mM Tris, 300 mM NaCl, 10% v/v glycerol, 20 mM imidazole, adjusted to pH 8 with HCl) and stored at −80 °C until further use. After thawing, they were resuspended in buffer BZ3 (2 ml per 1 g of pellet) supplemented with DNase I (10 μg/ml) and PMSF (100 μg/ml). After three passages through a French press at 16,000 psi, cell debris was removed by centrifugation at 30,000 × g for 30 min. The supernatant was then applied onto a 5 ml HisTrap HP column (GE Healthcare, USA) equilibrated with buffer BZ3. The column was washed with 10 column volumes (CV) of buffer BZ3, and the protein was eluted with a linear imidazole gradient (20–250 mM in buffer BZ3) at a flow rate of 1 ml/min. Fractions containing the protein at high purity were pooled and dialyzed against buffer B6 (50 mM HEPES, 50 mM NaCl, 5 mM MgCl₂, 0.5 mM EDTA, 10% v/v glycerol, adjusted to pH 7.2 with NaOH). Afterwards, the protein was concentrated in a centrifugal filter device (Amicon, USA), aliquoted, snap-frozen in liquid nitrogen and stored at −80 °C.

For use in lytic transglycosylase activity assays, MAS-DipM(26-609)-His₆ was subjected to an additional chromatographic step added after Ni-NTA affinity purification. To this end, the protein was dialyzed against buffer LS (50 mM Tris, 50 mM NaCl, adjusted to pH 8 with HCl) and applied to an HiTrap SP cation exchange column (GE Healthcare, USA) equilibrated with the same buffer. After elution with a linear NaCl

gradient (50–1000 mM in buffer LS), suitable fractions were pooled and dialyzed against buffer B6. Subsequently, the protein was concentrated, aliquoted, snap-frozen and stored at −80 °C.

**DipM^LytM.** His₆-SUMO-DipM(459-609) was overproduced in *E. coli* AI041 and purified by Ni-NTA affinity chromatography as described for full-length DipM. After elution from the HisTrap HP column, fractions containing the protein in high concentrations were pooled and dialyzed against buffer CB (50 mM Tris, 150 mM NaCl, 10% v/v glycerol, adjusted to pH 8 with HCl). The solution was then supplemented with 1 mM DTT and His₆-Ulp1 protease[88] at a 1:1000 molar ratio relative to His₆-SUMO-DipM(459-609) and incubated for 2 h at 4 °C. Subsequently, it was passed through a 0.22 μm filter to remove potential precipitates and applied onto a 5 ml HisTrap HP column (GE Healthcare, USA) equilibrated with buffer BZ3 as described above. Flow-through fractions containing DipM(459-609) at high purity were pooled and dialyzed against buffer B6. Finally, the protein was concentrated, aliquoted, snap-frozen and stored at −80 °C.

For crystallization, DipM(459-609) was further purified by size exclusion chromatography after removal of the SUMO tag. To this end, flow-through fractions from the second Ni-NTA affinity purification step containing the protein at high purity were concentrated and then applied onto a HiLoad 16/60 Superdex 75 prep grade column (GE Healthcare, USA) equilibrated with a buffer containing 50 mM HEPES and 100 mM NaCl, adjusted to pH 7.2 with NaOH. After elution at a flow rate of 0.5 ml/min, peak fractions were concentrated, aliquoted, snap-frozen and stored at −80 °C.

**SdpA.** To purify His₆-SdpA(21-699), cells of *E. coli* AI033 were grown in LB medium at 37 °C and then shifted to 18 °C prior to the induction of protein overproduction by the addition of 1 mM IPTG (at an OD₆₀₀ of 0.6). After incubation of the culture at 18 °C for another 18 h, the cells were harvested, washed with buffer BZ3 and stored at −80 °C. For further processing, the thawed cells were resuspended in buffer BZ3 (2 ml per 1 g of cell pellet) supplemented with DNase I (10 μg/ml) and PMSF (100 μg/ml). After three passages through a French press at 16,000 psi, cell debris was removed by centrifugation at 30,000 × g for 30 min. The supernatant was then applied onto a 5 ml HisTrap HP column (GE Healthcare, USA) equilibrated with buffer BZ3. The column was washed with 10 column volumes (CV) of buffer BZ3, and protein was eluted with a linear imidazole gradient (20–250 mM in buffer BZ3) at a flow rate of 1 ml/min. Fractions containing His₆-SdpA at high purity were pooled, concentrated and applied onto a HiLoad 16/60 Superdex 200 prep grade size exclusion chromatography columns equilibrated with buffer B6. The peak fractions were concentrated, aliquoted, snap-frozen and stored at −80 °C.

**SdpB.** His₆-SUMO-SdpB(26-536) was overproduced in *E. coli* AI041 and purified by Ni-NTA chromatography as described for His₆-SdpA(21-699). After elution from the HisTrap HP column, fractions containing the protein in high concentrations were pooled and dialyzed against buffer CB. The solution was supplemented with 1 mM DTT and His6-Ulp1 protease and incubated for 2 h at 4 °C. Subsequently, it was passed through a 0.22 μm filter to remove potential precipitates and applied onto a 5 ml HisTrap HP column (GE Healthcare, USA) equilibrated with buffer BZ3. Flow-through fractions containing the protein at high purity were pooled and dialyzed against buffer B6. Finally, the protein was concentrated, aliquoted, snap-frozen and stored at −80 °C.

**AmiC.** His₆-SUMO-AmiC(35-395) was overproduced in *E. coli* AI061 and purified essentially as described for His₆-SUMO-SdpB(26-536). However, the flow-through fractions obtained after the second round of Ni-NTA chromatography on a HisTrap HP column were dialyzed against buffer B7 (50 mM HEPES, 300 mM NaCl, 5 mM MgCl₂, 0.5 mM EDTA,

10% v/v glycerol, adjusted to pH 7.2 with NaOH) prior to concentration, aliquotation and snap-freezing.

**FtsN^Peri.** His$_6$-SUMO-FtsN(51-266) was overproduced in *E. coli* AI060 and isolated as described for His$_6$-SUMO-DipM(459-609), with the addition of a size exclusion chromatography step to improve the purity of the preparation. After removal of the SUMO tag, fractions from the second Ni-NTA affinity purification step that were highly enriched in FtsN(51-266) were concentrated and applied to a HiLoad 16/60 Superdex 75 prep grade column (GE Healthcare, USA) equilibrated with buffer B6. Fractions containing the protein at high purity were pooled, concentrated, aliquoted, snap-frozen and stored at −80 °C.

**CrbA^SPOR.** His$_6$-SUMO-CrbA(371-451) was overproduced in *E. coli* AI075 and processed as described for His$_6$-SUMO-DipM(459-609), with the addition of a last step in which the protein was dialyzed against buffer LS and then applied onto a HiTrap SP cation exchange column (GE Healthcare, USA) equilibrated in the same buffer. CrbA(371-451) was eluted with a linear NaCl gradient (50–1000 mM in buffer LS). Fractions containing the protein at high purity were pooled and dialyzed against buffer B6. Subsequently, they were concentrated, aliquoted, snap-frozen and stored at −80 °C.

**LdtD.** His$_6$-SUMO-LdtD(26-502) was overproduced in *E. coli* MAB408 and purified essentially as described for His$_6$-SUMO-SdpB(26-536). However, after the second round of Ni-NTA chromatography on a HisTrap HP column, the fractions obtained were dialyzed against buffer B8 (50 mM HEPES, 150 mM NaCl, 5 mM MgCl$_2$, 0.5 mM EDTA, 10% v/v glycerol, adjusted to pH 7.2 with NaOH) prior to concentration, aliquotation and snap-freezing.

## Protein crystallization and structure determination

High-throughput crystallization trials were performed in 96-well plates at 18 °C in sitting drops consisting of 250 nl of protein solution (13.6 mg/ml) and 250 nl of precipitation solutions from commercial crystallization screens (Hampton Research, USA), using an Innovadine crystallization robot. Conditions producing crystals were optimized and scaled up to 1 μl of protein solution and 1 μl of precipitant against 150 μl of the crystallization solution in the reservoir. DipM crystallized in 0.02 M magnesium chloride, 0.1 M HEPES pH 7.5 and 22% polyacrylic acid sodium salt 5100. Crystals were cryoprotected in 30% polyethylene glycol prior to flash freezing. Diffraction data were collected at the ALBA synchrotron (XALOC beamline) using the Pilatus 6 M detector. DipM crystals belonged to the P22$_1$2$_1$ space group with the dimensions $a = 65.869$, $b = 105.839$, $c = 108.431$, $\alpha = \beta = \gamma = 90°$, and diffracted up to 2.25 Å resolution at a wavelength of 0.979 Å. The datasets were processed with XDS (version Jan 31, 2020)[89] and Aimless 0.7.4[90]. The asymmetric unit was formed by four monomers, with 56.09% of solvent content. Structure determination was performed by molecular replacement, using Zoocin A from *Streptococcus equi* (PDB: 5KVP; 46.67% of sequence identity with DipM) as a search model. The model obtained was then completed manually using Coot 0.8.9.2[91], followed by refinement using PHENIX 1.15.2[92] and REFMAC 5.8.0258[93]. A summary of the refinement statistics is given in Supplementary Table 1. Structures were visualized with ChimeraX 1.5[94].

## Co-immunoprecipitation analysis

To identify interactors of DipM, SdpA, AmiC, LdpF and CrbA, we used strains producing FLAG-tagged bait proteins or untagged versions thereof (as negative controls) under the control of the xylose-inducible P$_{xyl}$ promoter in place of the respective native proteins. In the case of FtsN, a strain bearing *gfp* fused the endogenous *ftsN* gene was employed, and the wild-type strain NA1000 was used as negative control. The cells were grown in 500 ml (for DipM and SdpA) or 200 ml

(for all remaining proteins) M2G medium, supplemented with 0.3% xylose when necessary, until they reached an OD$_{600}$ of 0.6. After the addition of formaldehyde to a final concentration of 0.6%, the cultures were incubated for 5 min at room temperature, before the crosslinking reaction was stopped by the addition of glycine to a final concentration of 125 mM. Cells were harvested by centrifugation (12,000 × *g*, 4 °C, 10 min), washed with wash buffer (50 mM sodium phosphate pH 7.4, 5 mM MgCl$_2$), pelleted and stored at −80 °C. For further processing, the pellets were resuspended in 10 ml of Co-IP buffer (20 mM HEPES pH 7.4, 100 mM NaCl, 20 % [v/v] glycerol, 10 mg/ml lysozyme, 5 μg/ml DNase I, 100 μg/ml PMSF) containing 0.5% (for DipM and SdpA) or 1% (remaining proteins) Triton X-100. The cells were then incubated for 30 min on ice and disrupted by three passages through a French press. For each sample, the lysate was centrifuged (13,000 × *g*, 4 °C, 5 min) to remove intact cells and cell debris, and the supernatant was incubated with anti-FLAG M2 Magnetic Beads (Sigma-Aldrich, Germany; Cat. #: M8823) or GFP-Trap Magnetic Agarose (ChromoTek, Germany; Cat. #: gtma) for 2 h at 4 °C in a rotator. The beads were then collected by centrifugation (4000 × *g*, 4 °C), resuspended in 700 μl of 100 mM ammoniumbicarbonate, agitated vigorously and washed three times in 100 mM ammoniumbicarbonate using a magnetic separator. After removal of the supernatant of the last wash, the beads were resuspended in 100 μl of elution buffer 1 (1.6 M urea, 100 mM ammoniumbicarbonate, 5 μg/ml trypsin) and incubated for 30 min in a thermomixer (27 °C, 1200 rpm). After collection of the beads in a magnetic separator, the supernatant was transferred to a new tube. The beads were resuspended in 40 μl of elution buffer 2 (1.6 M urea, 100 mM ammoniumbicarbonate, 1 mM tris[2-carboxyethyl]phosphine) and collected again. Subsequently, the supernatant was combined with the previous eluate, and the elution with elution buffer 2 was repeated one more time. The pooled fractions were left overnight at room temperature. On the following day, 40 μl of iodoacetamide (5 mg/ml) were added, and the mixture was incubated for 30 min in the dark. After the addition of 150 μl of 5% [v/v] trifluoroacetic acid (TFA), the mix was passed through a C-18 microspin column (Harvard Apparatus), previously conditioned with acetonitrile and equilibrated with buffer A (0.1% [v/v] TFA in water). The column was then washed three times with 150 μl of buffer C (5% [v/v] acetonitrile, 95% [v/v] water and 0.1% [v/v] TFA). The peptides were eluted by washing the column three times with 100 μl of buffer B (50% [v/v] acetonitrile, 49.9% water [v/v] and 0.1% [v/v] TFA). The combined eluates were then dried under vacuum, and peptides were suspended in LC buffer (0.15% [v/v] formic acid, 2% [v/v] acetonitrile) by 20 pulses of ultrasonication (amplitude 20, cycle 0.5) and shaking for 5 min at 1400 rpm and 25 °C.

LC-MS analysis of the peptide samples was carried out on a Q-Exactive Plus instrument connected to an Ultimate 3000 RSLC nano and a nanospray flex ion source (all Thermo Scientific). Peptide separation was performed on a reverse phase HPLC column (75 μm × 42 cm) packed in-house with C18 resin (2.4 μm, Dr. Maisch). The peptides were loaded onto a PepMap 100 precolumn (Thermo Scientific) and then eluted with a linear acetonitrile gradient (2–35% solvent B) over 60 or 90 min (solvent A: 0.15% [v/v] formic acid in water, solvent B: 0.15% formic acid [v/v] in acetonitrile). The flow rate was set to 300 nl/min. The spray voltage was set to 2.5 kV, and the temperature of the heated capillary was set to 300 °C. Survey full-scan MS spectra (*m/z* = 375–1500) were acquired in the Orbitrap with a resolution of 70,000 full width at half maximum at a theoretical *m/z* 200 after accumulation of 3 × 106 ions in the Orbitrap. Based on the survey scan, up to 10 of the most intense ions were subjected to fragmentation using high collision dissociation (HCD) at 27% normalized collision energy. Fragment spectra were acquired at 17,500 resolution. The ion accumulation time was set to 50 ms for both MS survey and MS/MS scans. To increase the efficiency of MS/MS attempts, the charged state screening modus was enabled to exclude unassigned and singly charged ions. The dynamic exclusion duration was set to 30 s.

The resulting raw data were analyzed using Mascot 2.5 (Matrix Science, USA). Search results were loaded into Scaffold 4 (Proteome Software) to extract total spectrum counts for further analysis. The peptide count data were loaded in Perseus 1.5.8.5[95] to generate volcano plots. In brief, one unit was added to all the counts to eliminate the zeroes and then log2 was applied to all data. Columns were classified according to whether they belonged to the sample or negative control, and volcano plots were generated using the default settings. The resulting data on enrichment (difference to negative control) and significance ($-\log_{10}$ of $p$ value) were exported to Microsoft Excel 2019 (Supplementary Data 1), where they were re-plotted to generate the figures.

## Biolayer interferometry

Bio-layer interferometry experiments were conducted in a BLItz system (PALL Life sciences, USA) equipped with High Precision Streptavidin (SAX) biosensors (PALL Life sciences, USA). Proteins were biotinylated using NHS-PEG4-Biotin (Thermo Scientific, USA) following the manufacturer's recommendations. After immobilization on the sensor surface and the establishment of a stable baseline, the biotinylated proteins were probed with the indicated analytes. After the binding step, the sensors were transferred to an analyte-free buffer to follow the dissociation kinetics. The extent of non-specific interactions was determined by analyzing the binding of the respective analyte to unmodified sensors. All BLI analyses were performed in a buffer containing 50 mM HEPES/NaOH pH 7.2, 50 mM NaCl, 5 mM MgCl$_2$, 0.1 mM EDTA, 10% glycerol, 10 µM BSA, 0.01% Triton X-100.

For analysis, BLI data were processed in Microsoft Excel 2019 using the Solver add-in. In brief, the association/dissociation curves were normalized against the baselines obtained prior to exposure of the sensor to the analyte. The wavelength shift values reached after equilibration of the binding reactions were fitted to a one-site binding model to calculate the equilibrium dissociation constants ($K_D$).

## Peptidoglycan digestion assays

To assay for lytic transglycosylase activity, SdpA (5 µM) alone or in combination with DipM or DipM$^{LytM}$ (5 µM) was incubated with purified peptidoglycan (0.5 mg/ml) from *E. coli* MC1061[96] in 20 mM HEPES/NaOH pH 7.5, 20 mM NaCl, 1 mM MgCl$_2$ in a total volume of 50 µl for 4 h in a thermal shaker set to at 37 °C and 900 rpm. In parallel, a control reaction containing no SdpA was performed. The reactions were stopped by heating the samples for 10 min at 100 °C in a dry-bed heat block. The samples were centrifuged at room temperature for 15 min at 16,000 × *g*. The supernatant was recovered and the pH adjusted to 3.5–4.5 with 20% phosphoric acid. The samples were analyzed by HPLC as published previously[97] but with a modified buffer B and a linear gradient over 140 min from buffer A (50 mM sodium phosphate pH 4.31 with 1 mg/l sodium azide) to buffer B (75 mM sodium phosphate pH 4.95, 30% methanol). Eluted muropeptides were detected by their absorbance at 205 nm. An overnight digestion of peptidoglycan with the soluble lytic transglycosylase Slt from *E. coli*[65] (5 µM) served as a positive control. The effect of DipM or DipM$^{LytM}$ on the activity of Slt was assayed as described for SdpA, but the samples were only incubated for 30 min prior to the analysis of the reaction products, unless indicated otherwise. Peaks were assigned by their retention time and quantified by UV absorbance. Statistical tests comparing the integrated peak areas were performed in Microsoft Excel 2019.

To assay for amidase activity, peptidoglycan (0.5 mg/ml) from *E. coli* MC1061 was incubated with AmiC alone and with AmiC combined with either DipM or DipM$^{LytM}$ (5 µM each) in 20 mM Hepes/NaOH pH 7.5, 20 mM NaCl in a total volume of 50 µl for 16 h in a thermal shaker set to 37 °C and 900 rpm. Control reactions did not contain AmiC. The samples were heated at 100 °C for 10 min in a dry-bed heat block, acidified to pH 4.8 and incubated further with 10 µg of cellosyl

(Hoechst, Frankfurt am Main, Germany) for 16 h at 37 °C in a thermal shaker. Subsequently, the samples were again heated at 100 °C for 10 min in a dry-bed heat block and centrifuged at room temperature for 15 min at 16,000 × *g*. After retrieval of the supernatant, the muropeptides in the supernatant were reduced with sodium borohydride and separated by HPLC analysis as described previously[97].

## Statistics and reproducibility

Statistical analyses were performed with Microsoft Excel 2019 or R 3.3.0+[82]. Unless indicated otherwise, all experiments were conducted at least three times independently, with similar results.

## Reporting summary

Further information on research design is available in the Nature Portfolio Reporting Summary linked to this article.

## Data availability

The atomic coordinates for the crystal structure of DipM$^{LytM}$ were deposited in the RCSB Protein Data Bank with the accession number 7QRL. Proteomics data have been deposited to the ProteomeXchange Consortium via the PRIDE partner repository with the dataset identifier PXD042464. Single-molecule tracking data are available from Figshare (https://doi.org/10.6084/m9.figshare.23140343.v1). All other data generated in this study are included in the manuscript, the Supplementary Material and the Source Data file. Source data are provided with this paper.

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

## Acknowledgements

We thank Aleksandra Zielińska for help in the initial phase of the project, Manuel Osorio-Valeriano for advice on protein purification and biochemistry, Julia Rosum and Olga Ebers for excellent technical

assistance, Daniela Vollmer for the purification of peptidoglycan, and the staff of the ALBA Synchrotron facility for support during crystallographic data collection. This work was supported by the University of Marburg (core funding to P.L.G. and M.T.), the Max Planck Society (Max Planck Fellowship to M.T.), the German Research Foundation (DFG; project 269423233—TRR 174 to P.L.G.), the United Kingdom Research and Innovation (UKRI) Strategic Priorities Fund (grant EP/T002778/1 to W.V.), the Spanish Agency of Research at the Ministry of Science and Innovation (grant PID2020-115331GB-I00 to J.A.H.) and the Swiss National Science Foundation (grant CRSII5_198737/1 to J.A.H.). A.I.-M. was a fellow of the International Max Planck Research School for Environmental, Cellular and Molecular Microbiology (IMPRS-Mic).

## Author contributions

A.I.-M. and M.T. conceived the study. A.I.-M. and M.B. constructed plasmids and strains, purified proteins and performed the BLI experiments. A.I.-M. performed the Co-IP analysis. M.B. conducted the growth analyses and subcellular localization studies. V.M.-R. and M.T.B. performed the crystallization screens, solved the crystal structure of DipM$^{LytM}$ and performed the molecular modeling studies. R.H.-T. conducted the single-particle tracking analysis. J.B. performed the peptidoglycan digestion assays. T.G. performed the mass spectrometric analyses. A.I.-M., M.B., V.M.-R., R.H.-T., P.R., J.B., M.T.B., T.G., W.V., J.A.H. and M.T. analyzed the data. W.V., P.L.G., J.A.H. and M.T. supervised the study. W.V., P.L.G., J.A.H. and M.T. secured funding. A.I.-M. and M.T. wrote the paper, with input from all other authors.

## Funding

## Competing interests

The authors declare no competing interests.
