## [Peer Review File · Nature Communications]

DipM controls multiple autolysins and mediates a regulatory feedback loop promoting cell constriction in *Caulobacter crescentus*Reviewer #1 (Remarks to the Author):

Izquierdo-Martinez and colleagues present a solid study of the DipM autolysin activator. Co-IP experiments identify several DipM interacting proteins which are then confirmed by reciprocal pull downs and Bilayer interferometry. The authors also show DipM can stimulate enzymatic activity of SdpA (a lytic transglycosylase) and AmiC (a peptidoglycan amidase), and characterise the LytM subdomain of DipM using x-ray crystallography. An alphafold multimer prediction for how the DipM lytM domain might interact with the AmiC is also presented. Finally the roles of DipM are discussed with relation to cell division and cell envelope integrity.

The overall thrust of the paper is that DipM has a multiple interactions that regulate key enzymes in the cell envelope of *C. crescentus*.

The work will be of interest to those studying cell division in *Caulobacter crescentus* and those with a wider interest in how cell envelope modifying enzymes are regulated.

The methods appear sound and their description should be sufficient to allow replication.

The manuscript is well-written and the figures are of good quality.

Numbered comments/questions/suggestions appear below.

1) Given the emphasis on DipM having multiple interaction partners, would it be useful to present alphafold multimer models DipM with partners identified here? (FtsN, SdpA, SdpB, CrbA etc)?

2) Line 305 states "DipM does not recruit its two targets through direct physical interaction but rather indirectly through the regulatory activity of its lytM domain during cell division" - but the Bilayer Interferometry presented in Figure 2 shows there is a direct interaction between DipM and SdpA/SdpB. Work in Fig 4 also suggests a direct interaction. Is there a discrepancy between the microscopy and the in vitro work? Is the interaction direct or not?

3) The alphafold prediction for the interaction between the DipM lytM domain and amiC (Fig 7) needs to be better validated. Authors find a DipM mutation (R589A) located at the predicted interface that lacks DipM activity in complementation studies (Figure 7E) - but there is no additional evidence to show that the mutation works by disrupting the interaction of DipM with its partners as suggested by the Alphafold model.

There may also be some concern as to whether the DipM-R589A mutant is stable - the expression level of the mutant is clearly lower than the wild type (as seen in the fluorescence image of 7E).

The authors need to show: (a) that the DipM-R589A protein is stable and (b), that R589A substitution is affecting the ability of the protein to bind partners such as amiC or SpdA etc.

One suggestion is to purify the DipM-R589A mutant and use Bilayer Interferometry to measure the affinity for AmiC (and/or SpdA). This will address the issue of mutant stability and test the mechanism by which the mutation is presumed to operate. This should be straightforward given interaction of the WT has already been measured in this manner

4) The description of the alphafold complex is possibly too in-depth given this is a prediction rather than an experimentally-determined structure. Lines 275-286 could be shortened.

5) Lines 294-296 currently seems to suggest that amidases binding to a groove in LytM is a new idea from the current work that could be applied to EnvC. But the binding of Amidases to the EnvC LytM groove is well-known (Peters 2013 Mol Micro & Cook 2020 PNAS). Lines 264-265 and 294-296 needs to be edited to better reflect the current state of the literature.

6) The discussion touches on the interaction of the EnvC with FtsX (line 357-359) - A reference to the FtsX/EnvC co-structure might be appropriate.

7) Finally, in *E. coli*, DolP has recently been found as an interaction partner with various amidases (Boelter 2022 Microbiology) - were any DolP-like homologues pulled out in the Co-IPs performed here in *C. crescentus*? Would it be possible to include access to the co-ip data in the supplemental information?

Reviewer #2 (Remarks to the Author):

In this manuscript, the authors explore the function and regulation of autolysins in *Caulobacter crescentus* with a particular focus on a fairly enigmatic, but important, player in morphogenesis called DipM. Using a satisfying diversity of approaches, they demonstrate that DipM interacts with at least five other proteins and explore the dynamics of these interactions using purified proteins in vitro and imaging in cells. They use functional in vitro assays to show that DipM stimulates the lytic transglycosylase activity of two proteins (SdpA and SdpB), and use genetics, imaging and single molecule tracking to explore the spatial relationship among these factors. In addition, the authors probe the interaction between the LytM domain of DipM and the amidase AmiC using structural approaches and modeling. Altogether, this comprehensive and rigorous study provides significant detail to illuminate the web of physical and regulatory interactions governing cell wall hydrolases in *Caulobacter* and will be of broad interest to those studying bacterial morphogenesis, division, and cell wall metabolism.

Specific (minor) comments for improvement:

1. Line 142 - "Fig. 1G" - spell the word "figure"
2. Line 191 - "Using this system, we found that none of the LysM-less variants was able to restore normal cell division in DipM depleted cells". It is hard to assess how well DipM domains complement just from the images Fig 5B and quantitation in 5C (and S3). Quantitative analysis of growth (via growth curves and/or spotting assays) would help demonstrate fitness of each strain.
3. Line 193 - "Figure B-D" add number "5"
4. Line 216 - after "LytM" include the word "domain" or "region"
5. Fig 5B legend - Scale bar = ?
6. Fig 5C - indicate in the legend what the cell length numbers represent - is it the mean of the three biological replicate means?
7. Fig 7E - Western blots should be included to ensure that the point mutants are produced at similar levels as WT. As for point 2 above, inclusion of a growth assay would be useful in assessing function of the mutants. Also, the legend indicates "(I)" but it should be (E).
8. Fig 8 legend - Scale bar = ?
9. Fig 9 legend, the second "(B)" should be "(C)".

Reviewer #3 (Remarks to the Author):

Cell division in bacteria involves the localized synthesis of the peptidoglycan (PG) cell wall to eventually form the daughter cell poles. The so-called septal PG made by the division machinery is initially shared between daughters and must be processed to promote daughter cell separation. This remodeling of the PG layer requires careful regulation of cell wall cleaving enzymes and the coordination of their activities with those of the cell wall synthesis machinery. In the model organism *E. coli*, proteins with degenerate/defective LytM domains were identified as important regulators of septal PG processing by cell wall cleaving enzymes called amidases. Additionally, cell wall processing by the amidases has been implicated in what is thought to be a feedback loop mechanism in which cell wall synthesis and processing at the division site reinforce each other to drive cell constriction. Although this paradigm is well established in *E. coli*, it remains unclear whether other bacteria also use this mechanism and/or how it might be modified to achieve different growth modes.

This paper from Izquierdo-Martinez and co-workers reports an investigation into the function of the LytM domain protein DipM from *Caulobacter crescentus* and how it promotes cell division. Understanding how DipM works is important because past work indicates that its function may have diverged somewhat from that of the LytM regulators in *E. coli*. Therefore, learning more about the similarities and differences between the two systems promises to reveal new insights into the underlying regulatory mechanisms controlling PG remodeling in a broad swath of bacteria.

The investigation started with a proteomic analysis of DipM interaction partners (along with partners of the related LytM protein LdpF). The analysis revealed that DipM has many interaction partners, and a subset were chosen for further study. Of interest were the interactions with the cell wall processing enzymes: SdpA and SdpB with lytic transglycosylase (LTase) activity, CrbA with carboxypeptidase activity, and the amidase AmiC. These proteins were previously implicated in pathways involving DipM. DipM is required for the recruitment of SdpA and SdpB to division sites, it stimulates AmiC activity *in vitro*, and it has been shown to be recruited to the cell poles with SdpA and CrbA to build stalks. BLI studies were performed to test the directness of the DipM interactions identified in the pull-downs. Also, the ability of DipM to activate SdpA and AmiC was tested *in vitro*. The structure of the LytM domain of DipM was solved and shown to have properties analogous to the *E. coli* EnvC protein. Also, the DipM-AmiC structure was modeled and probed with mutagenesis. Finally, the single molecule dynamics of DipM in cells were monitored and shown to be significantly altered in cells defective for the SdpA and B proteins. Based on the DipM interaction network and these data, the authors propose that DipM is involved in two feedback loops driving division.

The paper is well written and contains a wealth of interesting new data. However, support for the model must be strengthened by testing of the physiological relevance of the protein-protein interactions and *in vitro* enzyme assays provided in the report. Such experiments should be relatively straightforward given the structural information available (or that can be predicted by AlphaFold) and the genetic systems in place in the labs of the authors.

Major points to be addressed:

1) A major point of the paper is that DipM interacts with two different PG degrading enzymes to regulate their activity at the division site. However, there are no data showing that these interactions are important for DipM's cell division function in cells. As part of the paper, the authors model the AmiC-DipM interaction and make mutations in the interaction site predicted to disrupt the interaction. One of the mutants (R589A) is shown to be defective for cell division. But, it remains unclear whether the mutant is (a) stable in cells, (b) actually defective for interacting with AmiC, and (c) fails to activate AmiC activity *in vitro*. Adding this data would significantly strengthen the argument that part of DipM's function in cells is to activate AmiC. I think a similar line of investigation is required to test the role of DipM in SdpA activation. Presumably, a similar AlphaFold modeling could be performed to predict the SdpA-DipM interface and allow the construction of the appropriate mutants to test the physiological relevance of the interaction and *in vitro* activation. Without such data, I don't think the overall model is all that compelling, especially considering some of my other concerns below.

2) The proteomic protein-protein interaction data is not very convincing in its present form. Many proteins are identified as co-purifying with DipM. I realize that all hits cannot be followed up or validated from such an experiment, but many of the hits were much more significantly enriched than those division proteins chosen by the authors. The data seem cherry-picked to focus only on proteins that were already of interest from other experiments. I am therefore concerned about the specificity of the interactions observed. The results could either mean that DipM has a lot of interaction partners or that the purification is inherently dirty. Given that DipM is a cell wall binding protein, the assay may just pull down a lot of cell wall associated proteins from partially digested wall fragments in the extract. The reciprocal pull-down data has the same issues. DipM is one of many hits for each of the partners tested and it is not close to being the most significant hit for any of them. I therefore find it difficult to put a lot of stock in the pull-down data without some additional controls for specificity.

3) The BLI interaction studies also lack controls. Given the very weak affinities measured for

several of the partners, I am concerned about the specificity of the interactions observed with this assay. All proteins tested interact with DipM, which is concerning. Are there any proteins that do not interact with DipM in the BLI assay? Some negative control proteins are needed. Also, the DipM mutants predicted to be defective in interaction with partners based on modeling should be tested. Such data would both serve to boost confidence in the BLI assay and in the AlphaFold models.

4) The activation of SdpA by DipM seems pretty weak to me. The data could be interpreted as DipM activating SdpA by a direct interaction. However, it could also be that DipM binding to the cell wall modifies its structure in a way that makes it more accessible to cleavage by SdpA. I would therefore like to see controls with an unrelated LT enzyme to see if its activity is also affected by DipM, and a DipM only control. Also, the data would be much more convincing with mutant controls discussed in point 1. A time-course would also provide a better sense of the activation as well.

5) The DipM-FtsN interaction centers prominently in the model and is the major connection supporting a role for DipM in the second of the two feedback loops. However, the interaction is not validated beyond the proteomic and BLI data for which I have raised concerns above. To include the interaction so prominently in the overall model and title of the paper, more controls are needed to provide convincing support for the interaction with FtsN and its physiological relevance.

Minor comments:

6) I do not think the term "LytM factors" is ideal since it does not differentiate between those with intact active sites and those with degenerate/defective active sites. A better designation should be used.

7) Line 193: Figure B-D?

We thank the three reviewers for the time and effort they invested in reviewing our manuscript and for their constructive criticism, which helped to significantly improve our paper.

We have performed a number of additional analyses to address the issues raised. Most importantly, we have

- generated AlphaFold-Multimer models for all DipM-target complexes, which suggest that the conserved groove in the LytM domain of DipM acts as an interaction site for all target proteins.
- constructed two additional DipM variants with exchanges in the predicted target binding site and shown that all of these exchanges abolish the function of DipM *in vivo* and its interaction with AmiC and SdpA *in vitro*, thereby confirming the AlphaFold predictions.
- performed a negative control experiment that validates the specificity of the BLI-based interaction analysis.
- performed a negative control experiment showing that the moderate stimulatory effect of DipM or DipM^{LytM} on the lytic transglycosylase activity of SdpA is specific to SdpA and not observed for the distantly related lytic transglycosylase Slt from *E. coli*.
- conducted grow analyses of all strains producing mutant DipM variants.
- performed additional Western blot analyses confirming the stability of all mutant constructs.
- performed additional experiments to further corroborate the link between FtsN and DipM.

Please see below for detailed responses to the issues raised.

During the revision process, we had to realize that the DipM depletion strain used as the basis for various complementation experiments carried a suppressor mutation that attenuated its phenotype in the absence of DipM. As a consequence, the phenotypes observed for the mutant DipM variants also appeared less severe. We have now reconstructed all strains that were affected (new strains MAB501-MAB504 and MAB512-MAB515; see Supplemental Information), reanalyzed their phenotypes and reinvestigated the localization patterns of the mutant DipM-sfmTurquoise2^{ox} fusions they produced. The results obtained for the reconstructed strains are now included in Figure 4 (DipM variants lacking the LysM or LytM domains), Figure 7 (DipM variants with amino acid exchanges in the LytM domain) and Supplementary Figure 7 (DipM variants with amino acid exchanges in the C-terminal DipM-specific loop). For the strains shown in Figures 4 and 7, the phenotypes observed are more severe than those reported previously. Most importantly, we now observe that cells producing DipM variants that lack the LysM domains have strong morphological defects, whereas the corresponding previously used strains only showed moderate changes in cell shape. These results provide further support for a model in which the accumulation of DipM at midcell, driven mostly by the peptidoglycan-binding activity of its LysM domains, is critical to localize and/or activate its target proteins at the cell division site.

REVIEWER COMMENTS

Reviewer #1

Izquierdo-Martinez and colleagues present a solid study of the DipM autolysin activator. Co-IP experiments identify several DipM interacting proteins which are then confirmed by reciprocal pull downs and Bilayer interferometry. The authors also show DipM can stimulate enzymatic activity of SdpA (a lytic transglycosylase) and AmiC (a peptidoglycan amidase), and characterise the LytM subdomain of DipM using x-ray crystallography. An AlphaFold multimer prediction for how the DipM LytM domain might interact with the AmiC is also presented. Finally the roles of DipM are discussed with relation to cell division and cell envelope integrity.

The overall thrust of the paper is that DipM has multiple interactions that regulate key enzymes in the cell envelope of *C. crescentus*. The work will be of interest to those studying cell division in *Caulobacter crescentus* and those with a wider interest in how cell envelope modifying enzymes are regulated. The

methods appear sound and their description should be sufficient to allow replication. The manuscript is well-written and the figures are of good quality. Numbered comments/questions/suggestions appear below.

1) Given the emphasis on DipM having multiple interaction partners, would it be useful to present AlphaFold multimer models DipM with partners identified here? (FtsN, SdpA, SdpB, CrbA etc)?

We have now generated models of the complexes that DipM^{LytM} forms with its various interaction partners using AlphaFold-Multimer. The results are presented in a new supplemental figure (Supplementary Figure 13). In all of the models, the interaction partners are predicted to interact with the groove of DipM^{LytM} and/or its surroundings. These findings strongly support the conclusion that the face of the LytM domain containing the degenerate catalytic site serves as an interaction hub for all target proteins of DipM, consistent with the finding that these proteins compete for binding to DipM (Supplementary Figure 2).

2) Line 305 states "DipM does not recruit its two targets through direct physical interaction but rather indirectly through the regulatory activity of its lytm domain during cell division" - but the Bilayer Interferometry presented in Figure 2 shows there is a direct interaction between DipM and SdpA/SdpB. Work in Fig 4 also suggests a direct interaction. Is there a discrepancy between the microscopy and the in vitro work? Is the interaction direct or not?

Our data indeed show that there is a direct interaction between SdpA/B and DipM *in vitro*. We have now performed a more careful quantification of the localization patterns of the two SLTs and included cells at all stages of the cell cycle in the demographic analysis. A comparison of the patterns observed in the *dipMΔ35-458* and the wild-type backgrounds clearly shows that the localization of SdpA-mCherry and SdpB-mCherry to the cell division site is severely impaired in cells producing the mutant DipM variant (new Supplementary Figure 17). Thus, direct protein-protein interactions between DipM and SdpA/B indeed appear to have a central role in the recruitment of the two SLTs to midcell.

3) The alphafold prediction for the interaction between the DipM lytM domain and amiC (Fig 7) needs to be better validated. Authors find a DipM mutation (R589A) located at the predicted interface that lacks DipM activity in complementation studies (Figure 7E) - but there is no additional evidence to show that the mutation works by disrupting the interaction of DipM with its partners as suggested by the AlphaFold model. There may also be some concern as to whether the DipM-R589A mutant is stable - the expression level of the mutant is clearly lower than the wild type (as seen in the fluorescence image of 7E). The authors need to show: (a) that the DipM-R589A protein is stable and (b), that R589A substitution is affecting the ability of the protein to bind partners such as amiC or SpdA etc.

One suggestion is to purify the DipM-R589A mutant and use Bilayer Interferometry to measure the affinity for AmiC (and/or SpdA). This will address the issue of mutant stability and test the mechanism by which the mutation is presumed to operate. This should be straightforward given interaction of the WT has already been measured in this manner

We agree with reviewer #1 that the relevance of the postulated interaction site was not very well supported in the first version of the manuscript.

(a) We have now performed additional Western blot analyses of cells producing the R589A variant in the presence (wild-type morphology) or in the absence (filamentous morphology) of the native DipM protein (see new Supplementary Figure 14). The results show that the mutant DipM-sfmTurquoise2^{ox} fusion accumulates to the same levels as the wild-type fusion when produced in the wild-type background, indicating that its synthesis and stability is not affected by the amino acid exchange. In DipM-depleted cells, the R589A variant accumulates to higher levels than the wild-type fusion, which likely is an indirect consequence of cell filamentation.

(b) As suggested by reviewer #1, we have now purified DipM-R589A and analyzed its interaction with AmiC and SdpA *in vitro*. In both cases, the R589A exchange largely or completely abolished the interaction between DipM and its two regulatory targets (new Figure 7d). Moreover, we have now generated additional mutant variants of DipM in which hydrophobic residues at the bottom of the binding groove were exchanged (L537S, L539S). These amino acid substitutions also severely impaired DipM function *in vivo* and abolished the interaction of DipM with AmiC and SdpA *in vitro*. Together, these new findings strongly support the notion that the groove in the LytM domain of DipM critically contributes to the interaction of DipM with its regulatory targets.

4) The description of the alphafold complex is possibly too in-depth given this is a prediction rather than an experimentally-determined structure. Lines 275-286 could be shortened.

We agree and have now shortened this part of the manuscript.

5) Lines 294-296 currently seems to suggest that amidases binding to a groove in LytM is a new idea from the current work that could be applied to EnvC. But the binding of Amidases to the EnvC LytM groove is well-known (Peters 2013 Mol Micro & Cook 2020 PNAS). Lines 264-265 and 294-296 needs to be edited to better reflect the current state of the literature.

It was not our intention to state that the interaction of amidases with the groove of regulatory LytM domains is a new idea and have now edited the text accordingly. However, we would like to point out that, although the role of the groove in the interaction of regulatory LytM domains with amidases is well-established, the precise regions bound in their corresponding amidases have not yet been identified. Our results open the possibility that, unlike commonly thought, regulatory LytM domains do not bind directly to the inhibitory helix 5 but potentially to the long helix 6 to induce a conformational change that removes helix 5 from the catalytic site. However, more work is required to test this hypothesis.

6) The discussion touches on the interaction of the EnvC with FtsX (line 357-359) - A reference to the FtsX/EnvC co-structure might be appropriate.

Done.

7) Finally, in *E. coli*, DolP has recently been found as an interaction partner with various amidases (Boelter 2022 Microbiology) - were any DolP-like homologues pulled out in the Co-IPs performed here in *C. crescentus*? Would it be possible to include access to the co-ip data in the supplemental information?

We have analyzed the proteome of *C. crescentus* for proteins with a domain structure similar to that of DolP but did not detect any. The Co-IP data are now provided as a supplemental file.

Reviewer #2

In this manuscript, the authors explore the function and regulation of autolysins in *Caulobacter crescentus* with a particular focus on a fairly enigmatic, but important, player in morphogenesis called DipM. Using a satisfying diversity of approaches, they demonstrate that DipM interacts with at least five other proteins and explore the dynamics of these interactions using purified proteins *in vitro* and imaging in cells. They use functional *in vitro* assays to show that DipM stimulates the lytic transglycosylase activity of two proteins (SdpA and SdpB), and use genetics, imaging and single molecule tracking to explore the spatial relationship among these factors. In addition, the authors probe the interaction between the LytM domain of DipM and the amidase AmiC using structural approaches and modeling. Altogether, this

comprehensive and rigorous study provides significant detail to illuminate the web of physical and regulatory interactions governing cell wall hydrolases in *Caulobacter* and will be of broad interest to those studying bacterial morphogenesis, division, and cell wall metabolism.

Specific (minor) comments for improvement:

1. Line 142 – “Fig. 1G” – spell the word “figure”

Done.

2. Line 191 – “Using this system, we found that none of the LysM-less variants was able to restore normal cell division in DipM depleted cells”. It is hard to assess how well DipM domains complement just from the images Fig 5B and quantitation in 5C (and S3). Quantitative analysis of growth (via growth curves and/or spotting assays) would help demonstrate fitness of each strain.

We have now included growth curves for all strains with obvious phenotypes to assess their level of fitness.

3. Line 193 – “Figure B-D” add number “5”

Done.

4. Line 216 – after “LytM” include the word “domain” or “region”

This statement refers to the protein LytM of *S. aureus*, which largely consists of a single LytM domain. Thus, in this case, the term “LytM” is correct.

5. Fig 5B legend – Scale bar = ?

Done.

6. Fig 5C – indicate in the legend what the cell length numbers represent – is it the mean of the three biological replicate means?

This figure has now changed significantly in the revised version of the manuscript, and the numbers have been omitted.

7. Fig 7E – Western blots should be included to ensure that the point mutants are produced at similar levels as WT. As for point 2 above, inclusion of a growth assay would be useful in assessing function of the mutants. Also, the legend indicates “(I)” but it should be (E).

See our response to comment #3 of reviewer #1. Western blots analyzing the levels and stability of the mutant proteins are now shown in the new Supplementary Figure 14. We have now included growth curves to determine the fitness of all mutant strains that showed obvious phenotypes. The legend has been corrected.

8. Fig 8 legend – Scale bar = ?

The scale bars are now defined in all figure legends.

9. Fig 9 legend, the second “(B)” should be “(C)”.

Done.

Reviewer #3

Cell division in bacteria involves the localized synthesis of the peptidoglycan (PG) cell wall to eventually form the daughter cell poles. The so-called septal PG made by the division machinery is initially shared between daughters and must be processed to promote daughter cell separation. This remodeling of the PG layer requires careful regulation of cell wall cleaving enzymes and the coordination of their activities with those of the cell wall synthesis machinery. In the model organism *E. coli*, proteins with degenerate/defective LytM domains were identified as important regulators of septal PG processing by cell wall cleaving enzymes called amidases. Additionally, cell wall processing by the amidases has been implicated in what is thought to be a feedback loop mechanism in which cell wall synthesis and processing at the division site reinforce each other to drive cell constriction. Although this paradigm is well established in *E. coli*, it remains unclear whether other bacteria also use this mechanism and/or how it might be modified to achieve different growth modes.

This paper from Izquierdo-Martinez and co-workers reports an investigation into the function of the LytM domain protein DipM from *Caulobacter crescentus* and how it promotes cell division. Understanding how DipM works is important because past work indicates that its function may have diverged somewhat from that of the LytM regulators in *E. coli*. Therefore, learning more about the similarities and differences between the two systems promises to reveal new insights into the underlying regulatory mechanisms controlling PG remodeling in a broad swath of bacteria.

The investigation started with a proteomic analysis of DipM interaction partners (along with partners of the related LytM protein LdpF). The analysis revealed that DipM has many interaction partners, and a subset were chosen for further study. Of interest were the interactions with the cell wall processing enzymes: SdpA and SdpB with lytic transglycosylase (LTase) activity, CrbA with carboxypeptidase activity, and the amidase AmiC. These proteins were previously implicated in pathways involving DipM. DipM is required for the recruitment of SdpA and SdpB to division sites, it stimulates AmiC activity *in vitro*, and it has been shown to be recruited to the cell poles with SdpA and CrbA to build stalks. BLI studies were performed to test the directness of the DipM interactions identified in the pull-downs. Also, the ability of DipM to activate SdpA and AmiC was tested *in vitro*. The structure of the LytM domain of DipM was solved and shown to have properties analogous to the *E. coli* EnvC protein. Also, the DipM-AmiC structure was modeled and probed with mutagenesis. Finally, the single molecule dynamics of DipM in cells were monitored and shown to be significantly altered in cells defective for the SdpA and B proteins. Based on the DipM interaction network and these data, the authors propose that DipM is involved in two feedback loops driving division.

The paper is well written and contains a wealth of interesting new data. However, support for the model must be strengthened by testing of the physiological relevance of the protein-protein interactions and *in vitro* enzyme assays provided in the report. Such experiments should be relatively straightforward given the structural information available (or that can be predicted by AlphaFold) and the genetic systems in place in the labs of the authors.

Major points to be addressed:

1) A major point of the paper is that DipM interacts with two different PG degrading enzymes to regulate their activity at the division site. However, there are no data showing that these interactions are important for DipM's cell division function in cells. As part of the paper, the authors model the AmiC-DipM interaction and make mutations in the interaction site predicted to disrupt the interaction. One of the mutants (R589A) is shown to be defective for cell division. But, it remains unclear whether the mutant is (a) stable in cells, (b) actually defective for interacting with AmiC, and (c) fails to activate AmiC activity in

vitro. Adding this data would significantly strengthen the argument that part of DipM's function in cells is to activate AmiC. I think a similar line of investigation is required to test the role of DipM in SdpA activation. Presumably, a similar AlphaFold modeling could be performed to predict the SdpA-DipM interface and allow the construction of the appropriate mutants to test the physiological relevance of the interaction and *in vitro* activation. Without such data, I don't think the overall model is all that compelling, especially considering some of my other concerns below.

We agree with reviewer #3 that the relevance of the putative binding site was poorly supported in the previous version of our manuscript. To address this issue, we have now analyzed a total of three mutant DipM variants with exchanges in the predicted interaction groove (L537S, L539S and R589A). Importantly, all of these variants show greatly impaired functionality *in vivo* and no longer interact with AmiC or SdpA *in vitro*, while their stability and accumulation are not affected (Figure 7). These results corroborate the notion that the conserved groove of DipM acts as a multi-purpose binding site that mediates the interaction of DipM with multiple target proteins. Moreover, they demonstrate that the failure to bind autolysins abolishes the function of DipM, strongly supporting the hypothesis that DipM serves to regulate the activity of AmiC, SdpA and potentially other autolysins.

We have now generated AlphaFold-Multimer models of all DipM-target complexes identified. Notably, all interactors are predicted to bind to the groove of DipM and the loops surrounding it, again supporting a central role of this region in the interaction of DipM with its regulatory targets (new Supplementary Figure 13). Given the similarity of the binding modes, the identification of mutations that specifically disrupt the interaction of DipM with a single target protein (such as AmiC or SdpA) is a difficult and time-consuming endeavor. Attempts to disentangle the individual effects of DipM on its target proteins are thus clearly beyond the scope of the present study.

2) The proteomic protein-protein interaction data is not very convincing in its present form. Many proteins are identified as co-purifying with DipM. I realize that all hits cannot be followed up or validated from such an experiment, but many of the hits were much more significantly enriched than those division proteins chosen by the authors. The data seem cherry-picked to focus only on proteins that were already of interest from other experiments. I am therefore concerned about the specificity of the interactions observed. The results could either mean that DipM has a lot of interaction partners or that the purification is inherently dirty. Given that DipM is a cell wall binding protein, the assay may just pull down a lot of cell wall associated proteins from partially digested wall fragments in the extract. The reciprocal pull-down data has the same issues. DipM is one of many hits for each of the partners tested and it is not close to being the most significant hit for any of them. I therefore find it difficult to put a lot of stock in the pull-down data without some additional controls for specificity.

When performed with a highly sensitive mass spectrometry equipment, Co-IP analyses commonly yield relatively long lists of co-purifying proteins. The relevance of the hits identified is determined by considering both the fold enrichment of peptides and the *p* value obtained by comparing the results of multiple independent experiments. The top hits included a functional diverse set of proteins, such as cell wall-related proteins, division components, proteins involved in protein biogenesis and folding, uncharacterized proteins, etc. Given that factors involved in cell division are typically part of large complexes, it is conceivable that many of these proteins interact, directly or indirectly, with the respective bait proteins. However, as mentioned in the text, we specifically focused on SdpA, SdpB, AmiC, CrbA and FtsN because previous genetic and localization studies (Möll et al, 2010; Zielinska et al, 2017) had suggested a functional connection between these proteins and DipM. Since Co-IP data do not allow any differentiation between direct and indirect interactions, we additionally performed *in vitro* binding studies with purified proteins, which verified a direct mode of interaction in all cases.

The chance that proteins co-purify because they are pulled down with cell wall fragments is very low. During the ColP experiments, cells were extensively treated with lysozyme and cell debris, including large cell wall fragments, was removed before the start of the Co-IP procedure. Formally, we cannot rule out a low level of peptidoglycan contamination, but the data obtained suggest that it is not a significant source for false-positive hits. For instance, SdpB is highly enriched with DipM, but not with SdpA or other proteins as a bait, and similar interaction dependencies are observed in the reciprocal Co-IP experiments. If the majority of proteins were copurified with peptidoglycan fragments, the lists of co-purifying proteins should be very similar for all experiments performed. However, this is clearly not the case.

3) The BLI interaction studies also lack controls. Given the very weak affinities measured for several of the partners, I am concerned about the specificity of the interactions observed with this assay. All proteins tested interact with DipM, which is concerning. Are there any proteins that do not interact with DipM in the BLI assay? Some negative control proteins are needed. Also, the DipM mutants predicted to be defective in interaction with partners based on modeling should be tested. Such data would both serve to boost confidence in the BLI assay and in the AlphaFold models.

For all BLI experiments, we initially tested for non-specific interactions of the analytes with non-functionalized BLI sensors and only used conditions, in which such interactions were not detectable. Moreover, non-specific interactions with the sensors are not easily saturated at the protein concentrations used and would give rise to a linear increase in the wavelength shift with increasing analyte concentrations rather than to hyperbolic titration curves. The fact that we do see a saturation of the binding reaction at moderate protein concentrations reflects a limited number of binding sites on the immobilized protein and, thus, a specific interaction between ligand and analyte. The low micromolar affinities obtained for the different interactions are typical for dynamic systems, such as the cell division apparatus. Notably, the K_d values for the interactions of *E. coli* AmiC with its cognate LytM regulator NlpD and of *Anabaena sp.* AmiC1 and its LytM regulator Alr3353 were determined to be $\sim 15 \mu\text{M}$ (Rocaboy et al, 2013) and $43 \mu\text{M}$ (Bornikoel et al, 2018), respectively. The values measured in our work are in the same range, which supports the validity of the BLI measurements.

To further corroborate our data, we have now included a BLI analysis of the interaction between DipM and the periplasmic L,D-transpeptidase LdtD from *C. crescentus* (CCNA_01579), which was detected in the Co-IP analysis with DipM as a bait but not classified as a significant hit. As expected, we did not observe an interaction between the two proteins, which shows that DipM is not just “sticky” protein that binds other proteins without specificity.

In addition to the R589A variant, we have now generated two more mutant variants of DipM (L537S, L539S) in which we exchanged highly conserved hydrophobic residues in the groove of DipM that are predicted to be involved in the interaction with each of the target proteins (Supplementary Figure 13). All three variants showed strong functional defects *in vivo* (Figure 7a-c). We then tested the interaction of the R589A and L539S variants with AmiC and SdpA as representative targets *in vitro*. Importantly, both variants showed a severe reduction in their binding affinities (Figure 7d), which underscores the importance of the groove around the degenerate catalytic site as a specific binding interface for DipM targets.

4) The activation of SdpA by DipM seems pretty weak to me. The data could be interpreted as DipM activating SdpA by a direct interaction. However, it could also be that DipM binding to the cell wall modifies its structure in a way that makes it more accessible to cleavage by SdpA. I would therefore like to see controls with an unrelated LT enzyme to see if its activity is also affected by DipM, and a DipM only control. Also, the data would be much more convincing with mutant controls discussed in point 1. A time-course would also provide a better sense of the activation as well.

Notably, full-length DipM and DipM^{LytM} (the truncated version lacking the PG-binding LysM domains) stimulate the activity of Slt to a similar extent. However, in contrast to the full-length protein, DipM^{LytM} does not bind PG with appreciable affinity (Möll et al, 2010), so that it is unlikely to influence SLT activity indirectly by affecting peptidoglycan structure.

Nevertheless, we have now tested the ability of DipM and DipM^{LytM} to stimulate the activity of Slt from *E. coli*, a lytic transglycosylase with the same SLT domain as SdpA. We did not observe any increase in Slt activity in the presence of these proteins (new Supplementary Figure 3), ruling out the possibility that DipM promotes the activity of Slt indirectly through its PG binding activity. These results also demonstrate that the DipM and DipM^{LytM} preparations used in our analyses do not contain any contaminating SLT activity.

The degree of stimulation observed for SdpA is indeed smaller than that for AmiC. However, we still observe a more than fourfold increase in the activity of SdpA towards crosslinked peptidoglycan in the presence of DipM and DipM^{LytM}, which together with the DipM-dependent recruitment of SdpA to midcell may potentially be sufficient to promote cell constriction to the extent necessary. Moreover, it is possible that the stimulatory effect measured *in vitro* does not accurately reflect the effect *in vivo*, because the two proteins may be part of larger complexes that augment their activities or SdpA may preferentially act on certain crosslinked peptidoglycan substrates that are formed only transiently during cell constriction.

Since SLT activity assays are time-consuming, we were not able to perform additional assays with mutant DipM variants or at higher time-resolution within the time-frame available. However, we now show that the R589A and L539S variants of DipM have lost their ability to interact with AmiC or SdpA *in vitro* (new Figure 7d) and to properly control cell division *in vivo* (Figure 7a-c). These results strongly support our model that the function of DipM depends on the interaction of its LytM domain with multiple autolysins and the localization/activation of these autolysins at the cell division site.

5) The DipM-FtsN interaction centers prominently in the model and is the major connection supporting a role for DipM in the second of the two feedback loops. However, the interaction is not validated beyond the proteomic and BLI data for which I have raised concerns above. To include the interaction so prominently in the overall model and title of the paper, more controls are needed to provide convincing support for the interaction with FtsN and its physiological relevance.

We agree that the experimental data we provide on the role of the direct interaction between DipM and FtsN is quite limited. Therefore, we have now changed the title and toned down our statements concerning the second feedback loop throughout the manuscript.

However, it is highly likely that this second feedback loop exists, although it may not be mainly driven by direct interactions between DipM and FtsN but rather by the effects of the two proteins on the structure of the septal peptidoglycan layer. On the one hand, DipM activates the activity of the amidase AmiC at the cell division site, leading to the formation of denuded peptidoglycan, which in turn acts as a landmark that is recognized by the SPOR domain of FtsN. On the other hand, FtsN is required for the recruitment of DipM to midcell (Möll et al, 2010), likely because FtsN-dependent peptidoglycan remodeling produces peptidoglycan structures that are specifically recognized by the LysM domains of DipM. Direct interactions between DipM and FtsN could help to promote this feedback loop and/or tune the activities of the two proteins dependent on their accumulation levels at midcell.

To further address this point, we have included new data showing that cells producing a SPOR-less, largely delocalized, but still partially functional FtsN variant (new Supplementary Figure 18) are no longer able to retain DipM at the division sites late in the cell cycle. This finding indicates that the physical accumulation of FtsN at the division site is key to stabilize DipM at this position in the late steps of the division process. As described above, we have now also included a negative control validating the direct interactions measured by BLI.

Considering the points outlined above, we believe it is still justified to show the second feedback loop in the final model and thus integrate the current knowledge about the localization/activation dependencies between the different components. However, we have now reduced the size of FtsN to put the focus on DipM and its regulatory targets SdpA and AmiC. In addition, we have now clearly indicated the indirect nature of the effects giving rise to the second feedback loop by dotted lines.

We would like to point out that in-depth analyses of the physiological relevance of the direct interaction between DipM and FtsN are not straightforward. The AlphaFold models suggest that FtsN, like other targets of DipM, interacts with the conserved groove in the LytM domain, so that it would be challenging (and potentially impossible) to identify mutations that specifically disrupt the interaction between DipM and FtsN while leaving other interactions intact. Identifying such mutations in FtsN is also a difficult endeavor because this protein is at the heart of a complex interaction network that also includes peptidoglycan synthases and regulatory division components. It would therefore be very challenging to discriminate effects induced by the lack of interaction with DipM from those induced by the lack of other interactions.

Minor comments:

6) I do not think the term “LytM factors” is ideal since it does not differentiate between those with intact active sites and those with degenerate/defective active sites. A better designation should be used.

The term “LytM factors” has been coined in the first publication characterizing the regulatory effect of enzymatically inactive LytM domains on AmiC activity in *E. coli* and has since then been widely used in the field. However, we agree that it is not ideal because it does not clearly differentiate between proteins with enzymatically active and regulatory LytM domains. Therefore, we have now replaced the term “LytM factors” by “LytM regulators” throughout the text.

7) Line 193: Figure B-D?

Done.

Reviewer #1 (Remarks to the Author):

The authors have addressed my concerns and the manuscript is much improved.

The new experiments dissecting the interaction between DipM and its binding partners (and the new controls showing expression of mutants) provide a much more solid footing for understanding of the role of DipM in *C. crescentus* division.

The authors are also to be commended for bringing the suppressor mutation in the DipM deletion strain to attention of reviewers at the front of the response-to-review and for reinvestigating in the suppressor-free background.

The work is interesting and well-evidenced and will likely prompt future investigation of interactions between DipM like proteins and various cell envelope factors.

Reviewer #2 (Remarks to the Author):

The authors have addressed all of our concerns with the initial submission. They have presented a rigorous and comprehensive study of DipM and its regulatory network in *Caulobacter* that will be of interest to the bacterial cell biology community.

Reviewer #3 (Remarks to the Author):

The authors have done an excellent job with the revision to address my concerns and those of the other reviewers. Congratulations on a great paper showing the diversity of regulatory targets of LytM proteins and the role of this regulation in cell division.